# IVC-Prune: Revealing the Implicit Visual Coordinates in LVLMs for Vision Token Pruning

**Zhichao Sun**[1,3] , **Yidong Ma**[2], **Gang Liu**[2], **Yibo Chen**[2], **Xu Tang**[2], **Yao Hu**[2], **Yongchao Xu**[1,3 ✉]

[1] School of Computer Science, Wuhan University    [2] Xiaohongshu Inc    [3] Hubei Luojia Laboratory

## Abstract

Large Vision-Language Models (LVLMs) achieve impressive performance across multiple tasks. A significant challenge, however, is their prohibitive inference cost when processing high-resolution visual inputs. While visual token pruning has emerged as a promising solution, existing methods that primarily focus on semantic relevance often discard tokens that are crucial for spatial reasoning. We address this gap through a novel insight into *how LVLMs process spatial reasoning*. Specifically, we reveal that LVLMs implicitly establish visual coordinate systems through Rotary Position Embeddings (RoPE), where specific token positions serve as **implicit visual coordinates** (IVC tokens) that are essential for spatial reasoning. Based on this insight, we propose **IVC-Prune**, a training-free, prompt-aware pruning strategy that retains both IVC tokens and semantically relevant foreground tokens. IVC tokens are identified by theoretically analyzing the mathematical properties of RoPE, targeting positions at which its rotation matrices approximate identity matrix or the $90°$ rotation matrix. Foreground tokens are identified through a robust two-stage process: semantic seed discovery followed by contextual refinement via value-vector similarity. Extensive evaluations across four representative LVLMs and twenty diverse benchmarks show that IVC-Prune reduces visual tokens by approximately 50% while maintaining $\geq 99\%$ of the original performance and even achieving improvements on several benchmarks.

## 1 Introduction

LVLMs achieve impressive performance in perception, understanding, and reasoning across a broad range of multimodal tasks. Rapid advances in both proprietary systems (*e.g.*, GPT-5, Gemini 2.5 Pro) and open-source families (*e.g.*, Qwen-VL Bai et al. (2025), InternVL Wang et al. (2025)) have enabled greater model capacity, extended context lengths, and high-resolution image processing. However, high-resolution images often generate thousands of visual tokens, leading to prohibitive memory usage and long inference latency. To mitigate these challenges, recent studies have focused on visual token pruning to remove redundant tokens while maximally preserving performance. Existing approaches can be broadly grouped into two categories: (1) Training-based methods that learn to aggregate or select tokens via architectural modifications Ye et al. (2025b); Shao et al. (2025). (2) Training-free methods that use attention scores or similarity metrics for token selection Ye et al. (2025a); Arif et al. (2025). While effective for general visual understanding, these methods suffer substantial performance drops on spatially sensitive tasks such as visual grounding and spatial reasoning. The issue arises because existing methods primarily focus on semantic relevance between text and visual tokens while overlooking spatially critical tokens. As illustrated in Fig. 1, preserving only semantically relevant "foreground" tokens causes performance drops in visual grounding tasks.

In this work, we investigate the mechanisms for spatial reasoning in LVLMs: how they perceive the **absolute locations of objects in arbitrary resolution images** using Rotary Position Embeddings (RoPE, widely adopted in current mainstream LVLMs). Our theoretical analysis shows that RoPE encodes relative positions between query and key tokens in self-attention. Crucially, when a key token's RoPE rotation matrix approximates either the identity matrix or a $90°$ rotation, self-attention isolates the *absolute* positional component of the query. This implies the existence of special token

---

✉Corresponding author: Yongchao Xu <yongchao.xu@whu.edu.cn>

*Code available at: https://github.com/FireRedTeam/IVC-Prune

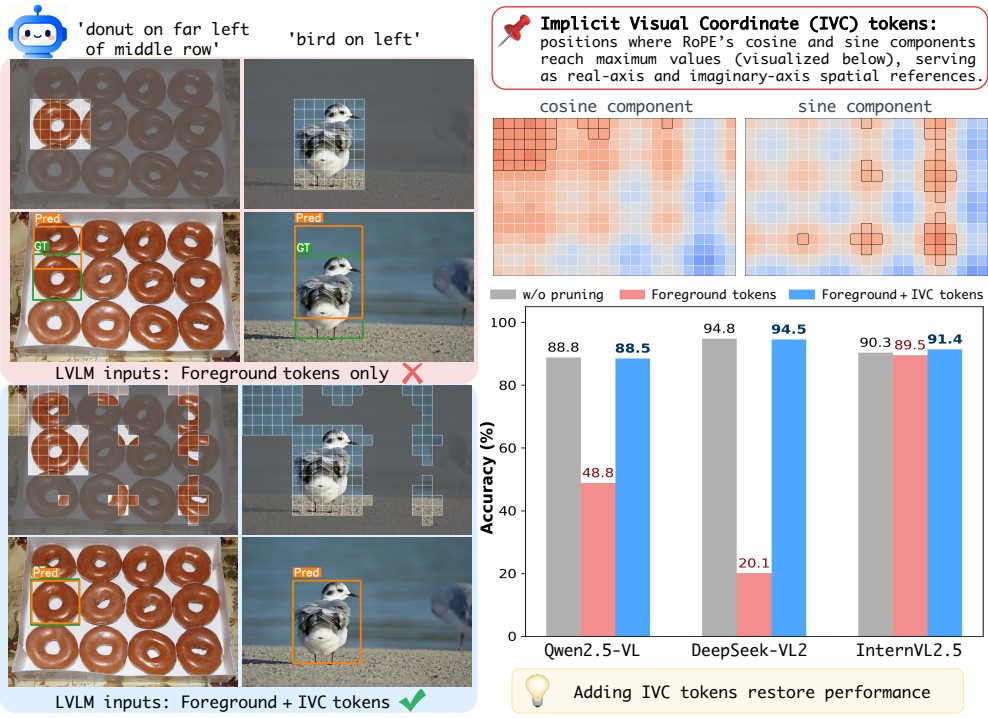

Figure 1: Implicit Visual Coordinate (IVC) tokens are crucial for spatial reasoning in LVLMs. **Left:** Visual grounding examples under different input settings. **Top right:** RoPE cosine and sine components across token positions, with IVC token locations (10% of total) marked in black. **Bottom right:** RefCOCO accuracy across three LVLMs under varying input settings, showing that adding IVC tokens largely restores performance. Detailed results and analysis are provided in Appendix A.4.

positions that act as spatial references: *real axis* (identity) and *imaginary axis* (90° rotation). These reference tokens form **implicit visual coordinates** (IVC), which are essential for spatial reasoning.

Building on this insight, we propose **IVC-Prune**, a training-free, prompt-aware pruning strategy that preserves both IVC tokens and semantically relevant foreground tokens. To identify IVC tokens, we rank the sum of cosine and sine components from RoPE. For robust foreground token selection across LVLM architectures, we employ a two-stage process: (1) Identify semantic seeds using value-vector similarity to mitigate positional bias in attention scores. (2) Leverage semantic seeds and text tokens to capture all relevant foreground tokens. Our experiments further reveal that sensitivity to early-layer pruning, reported in prior work, stems not from the pruning itself but from inadvertently removing IVC tokens. Based on this insight, we design a single-selection pruning strategy: the retained token set is determined once at a selected intermediate layer while preserving original position IDs. This selection is then applied to prune the KV caches in all earlier layers and is also used in later layers. This approach maximizes KV-cache reduction for efficient inference.

We evaluate IVC-Prune across four representative LVLMs (Qwen2.5-VL, InternVL 2.5, DeepSeek-VL2, and LLaVA v1.5) and twenty diverse benchmarks spanning visual grounding, reasoning, hallucination evaluation, and OCR tasks. Results show that IVC-Prune reduces visual tokens by approximately 50% while maintaining ≥99% of the original performance, and in some cases achieving performance improvements. Notably, on visual grounding tasks, IVC-Prune significantly outperforms existing pruning methods. Moreover, results show that IVC tokens can be seamlessly integrated into other pruning methods to consistently enhance their spatial reasoning capabilities. Our contributions are summarized as follows:

- To the best of our knowledge, we present the first theoretical analysis that LVLMs implicitly establish visual coordinate systems through RoPE's mathematical structure, providing novel insights into their spatial reasoning mechanisms.
- We propose IVC-Prune, a novel, training-free pruning strategy that preserves both IVC tokens and semantically relevant tokens, and introduces a robust two-stage selection process that is generalizable across LVLM architectures and benchmarks.

## 2 RELATED WORKS

**Large Vision-Language Models.** In recent years, Large Vision-Language Models (LVLMs) have emerged as a pivotal technology in artificial intelligence. Commercial models such as GPT-5 OpenAI (2025), Claude Sonnet 4 Claude (2025) and Gemini 2.5 Pro Comanici et al. (2025) demonstrate remarkable multimodal understanding and reasoning capabilities. In parallel, the open-source community has also rapidly advanced, starting with the pioneering LLaVA Liu et al. (2023). Subsequent advances include Qwen2.5-VL Bai et al. (2025), which supports native high-resolution inputs, and DeepSeek-VL2 Wu et al. (2024), which adopts a Mixture-of-Experts (MoE) architecture for greater parameter efficiency. Further advances include long-context reasoning capabilities in models like Kimi-VL Du et al. (2025) and the integration of reinforcement learning in InternVL 3.5 Wang et al. (2025). Despite these successes, the trend towards greater model scales, longer context processing, and higher input resolution has resulted in prohibitive computational costs. These costs have become a major bottleneck for deploying LVLMs in real-world, latency-sensitive applications.

**Token Pruning.** Token pruning reduces tokens in LLMs and LVLMs, lowering computational costs and improving efficiency. In LLMs, methods such as StreamingLLM Xiao et al. (2024) and MInference Jiang et al. (2024) retain attention sink tokens and local context tokens to support long-context. SepLLM Chen et al. (2025) enhances performance by also preserving separator tokens. In LVLMs, visual tokens typically far outnumber text tokens, making visual token pruning particularly important. Existing visual token pruning methods can be broadly classified into **training-based** and **training-free** approaches. Training-based approaches generally fall into two subcategories: (1) *Learnable query aggregation*, where models such as Qwen-VL Bai et al. (2024), MQT Hu et al. (2024), LLaMA-VID Li et al. (2024b), and VoCo-LLaMA Ye et al. (2025c) employ learnable queries to aggregate tokens in a manner similar to Q-Former Li et al. (2023b); (2) *Learned token selection*, where methods like LVPruning Sun et al. (2025) and DynamicLLaVA Huang et al. (2025) train modules to predict which tokens can be safely removed. However, these approaches often require substantial training costs and architectural modifications.

Training-free methods comprise: (1) *Clustering or merging*, such as LLava-PruMerge Shang et al. (2024), SparseVLM Zhang et al. (2025c), and PACT Dhouib et al. (2025), which group similar tokens to reduce redundancy and mitigate information loss. However, these methods often require rebuilding token position IDs, which can hurt performance Chien et al. (2025) on precise localization tasks such as visual grounding. (2) *Attention/similarity-based pruning*, including FastV Chen et al. (2024a), FlowCut Tong et al. (2025), TopV Yang et al. (2025), PDrop Xing et al. (2025), and SparseVILA Khaki et al. (2025) which use attention scores or similarity metrics to identify important tokens. However, these methods primarily focus on semantic relevance and often neglect spatially critical tokens, which may lead to drops in grounding accuracy. Motivated by this gap, we focus on visual grounding as a representative spatially sensitive task. Our theoretical analysis reveals that certain visual tokens implicitly act as spatial coordinates essential for spatial reasoning. Leveraging this insight, we develop a simple but effective pruning strategy that explicitly preserves these visual coordinate tokens alongside semantically relevant ones, yielding superior performance in both visual grounding and general benchmarks.

## 3 METHOD

### 3.1 BACKGROUND: ROTARY POSITION EMBEDDINGS IN ATTENTION

Rotary Position Embeddings (RoPE) Su et al. (2024), the mainstream positional encoding in LVLMs, encode positional information by applying structured rotations to feature vectors. For a $d$-dimensional vector $\boldsymbol{v}$, RoPE divides it into $d/2$ two-dimensional subspaces. Each pair $(\boldsymbol{v}_{2k}, \boldsymbol{v}_{2k+1})$, corresponding to a token at position $m$, is rotated as follows:

$$\begin{bmatrix} \boldsymbol{v}'_{2k} \\ \boldsymbol{v}'_{2k+1} \end{bmatrix} = \underbrace{\begin{pmatrix} \cos(m\theta_k) & -\sin(m\theta_k) \\ \sin(m\theta_k) & \cos(m\theta_k) \end{pmatrix}}_{\triangleq \boldsymbol{R}(m,\theta_k)} \begin{bmatrix} \boldsymbol{v}_{2k} \\ \boldsymbol{v}_{2k+1} \end{bmatrix}, \quad k = 0, \ldots, \frac{d}{2} - 1, \tag{1}$$

where $\theta_k = 10000^{-2k/d}$ is a predefined frequency for the $k$-th pair. Let $\mathbf{R}_m = \mathrm{diag}(\mathbf{R}(m,\theta_0), \ldots, \mathbf{R}(m,\theta_{d/2-1}))$ denote the block-diagonal matrix that applies these rotations to

the full $d$-dimensional vector. In a self-attention layer, for input features $\boldsymbol{x}_n$ and $\boldsymbol{x}_m$ at absolute positions $n$ and $m$, the queries and keys are computed as:

$$\boldsymbol{q}_n = \boldsymbol{R}_n(\boldsymbol{W}_q \boldsymbol{x}_n) \in \mathbb{R}^d, \quad \boldsymbol{k}_m = \boldsymbol{R}_m(\boldsymbol{W}_k \boldsymbol{x}_m) \in \mathbb{R}^d, \tag{2}$$

where $\boldsymbol{W}_q$ and $\boldsymbol{W}_k$ are the query and key projection matrices, respectively. The attention score is given by the dot product between the rotated queries and keys. Since $\boldsymbol{R}_n$ is an orthogonal rotation matrix, its transpose is equivalent to a rotation by the negative angle ($\boldsymbol{R}_n^\top = \boldsymbol{R}_{-n}$), we can rewrite:

$$\begin{aligned}
\text{AttentionScore}(\boldsymbol{q}_n, \boldsymbol{k}_m) &= (\boldsymbol{R}_n \boldsymbol{W}_q \boldsymbol{x}_n)^T (\boldsymbol{R}_m \boldsymbol{W}_k \boldsymbol{x}_m) \\
&= \boldsymbol{x}_n^T \boldsymbol{W}_q^T \boldsymbol{R}_{-n} \boldsymbol{R}_m \boldsymbol{W}_k \boldsymbol{x}_m \\
&= \boldsymbol{x}_n^T \boldsymbol{W}_q^T \boldsymbol{R}_{m-n} \boldsymbol{W}_k \boldsymbol{x}_m.
\end{aligned} \tag{3}$$

Eq. 3 reveals that the attention score inherently depends on the *relative* position $(m - n)$. This property provides RoPE with a natural mechanism for encoding relative positional relationships.

## 3.2 EXPLORING THE IMPLICIT VISUAL COORDINATE

While RoPE naturally encodes relative positions, tasks such as spatial reasoning require knowledge of absolute object positions in an image. This suggests the need for absolute reference coordinates. We hypothesize that models can implicitly establish such coordinates through RoPE's deterministic rotation matrices, whose periodic and orthogonal properties naturally define coordinate reference points. Consider the attention score: $\text{Score}(\boldsymbol{q}_n, \boldsymbol{k}_m) = \boldsymbol{x}_n^T \boldsymbol{W}_q^T \boldsymbol{R}_{-n} \boldsymbol{R}_m \boldsymbol{W}_k \boldsymbol{x}_m$. When the model attends to reference tokens at positions $m$, where the rotation matrix $\boldsymbol{R}_m$ approximates key canonical transformations (*e.g.*, identity matrix or $90°$ rotation matrices), the attention effectively isolates the query's absolute positional component $\boldsymbol{R}_n$. This motivates identifying positions $m$ whose rotation matrices serve as these canonical basis operators for an implicit coordinate system.

**Real-Axis Reference.** Based on our analysis, an ideal real-axis reference corresponds to the identity transformation. We search for positions $m$ where $\boldsymbol{R}_m$ is close to the identity matrix $\boldsymbol{I}$, as measured by the squared Frobenius norm:

$$\begin{aligned}
\|\boldsymbol{R}_m - \boldsymbol{I}\|_F^2 &= \sum_{k=0}^{d/2-1} \|\boldsymbol{R}(m, \theta_k) - \boldsymbol{I}_2\|_F^2 \\
&= \sum_{k=0}^{d/2-1} \left\| \begin{pmatrix} \cos(m\theta_k) - 1 & -\sin(m\theta_k) \\ \sin(m\theta_k) & \cos(m\theta_k) - 1 \end{pmatrix} \right\|_F^2 = \sum_{k=0}^{d/2-1} 4\big(1 - \cos(m\theta_k)\big), \quad (4)
\end{aligned}$$

where $\boldsymbol{I}_2 = \left(\begin{smallmatrix} 1 & 0 \\ 0 & 1 \end{smallmatrix}\right)$ and $\boldsymbol{I} = \text{diag}(\boldsymbol{I}_2, \ldots, \boldsymbol{I}_2)$. Minimizing this distance is equal to maximizing the sum of the cosine terms. Accordingly, we define the real-axis score for a position $m$ as:

$$V(m) = \sum_{k=0}^{d/2-1} \cos(m\theta_k), \tag{5}$$

which is equivalent to summing the **cosine components** of the positional embedding across all dimensions. Positions $m$ that maximize $V(m)$ are thus appropriate candidates for **real-axis references**.

**Imaginary-Axis Reference.** To complete the coordinate frame, an orthogonal axis is required. In each 2D feature subspace, this axis corresponds to a $90°$ counterclockwise rotation, represented by $\boldsymbol{J}_2 = \left(\begin{smallmatrix} 0 & -1 \\ 1 & 0 \end{smallmatrix}\right)$, which extends to higher dimensions as the block-diagonal matrix $\boldsymbol{J} = \text{diag}(\boldsymbol{J}_2, \ldots, \boldsymbol{J}_2)$. We identify positions $m$ whose rotation matrices $\boldsymbol{R}_m$ closely approximate $\boldsymbol{J}$. The distance is given by:

$$\|\boldsymbol{R}_m - \boldsymbol{J}\|_F^2 = \sum_{k=0}^{d/2-1} \|\boldsymbol{R}(m, \theta_k) - \boldsymbol{J}_2\|_F^2 = \sum_{k=0}^{d/2-1} 4\big(1 - \sin(m\theta_k)\big). \tag{6}$$

Minimizing this distance equals maximizing the sum of sines. We define the imaginary-axis score:

$$U(m) = \sum_{k=0}^{d/2-1} \sin(m\theta_k), \tag{7}$$

which aggregates the sine components of the positional embedding across all dimensions. Positions $m$ that maximize $U(m)$ serve as **imaginary-axis references**, providing a consistent $90°$ phase shift relative to the real-axis references and enabling the construction of a stable implicit coordinate system.

**Implications.** This analysis reveals that RoPE's mathematical properties naturally enable LVLMs to construct implicit coordinate systems. The functions $V(m)$ and $U(m)$ identify special positions that serve as coordinate anchors, thereby providing a mathematical foundation for absolute spatial reasoning in visual tasks. Importantly, these coordinate references emerge from the inherent periodicity and orthogonality properties of RoPE, suggesting that spatial understanding in LVLMs may be structured. This implicit coordinate system provides a theoretical basis for understanding how LVLMs perceive the absolute locations of objects in images of arbitrary resolution.

### 3.3 IMPLICIT VISUAL COORDINATE FOR TOKEN PRUNING

We propose a *training-free* token pruning strategy for LVLMs, applied within the language decoder to enable *prompt-aware* pruning. Our method specifically preserves two crucial visual token types:

- **Implicit Visual Coordinate (IVC) tokens**: Tokens that are essential for spatial reasoning.
- **Foreground tokens**: Visual tokens that are semantically aligned with the text prompt.

**IVC Token Selection.** Following the analysis in Section 3.2, we select IVC tokens by ranking each token position $m$ using the coordinate scores $V(m)$ and $U(m)$. We retain the top-$k_c$ tokens for each score and combine them to form the IVC token set:

$$\mathcal{I}_{\text{ivc}} = \arg \text{TopK}(\{V(m)\}, \ k_c) \ \cup \ \arg \text{TopK}(\{U(m)\}, \ k_c). \tag{8}$$

**Foreground Token Selection.** We employ a two-stage procedure to identify foreground tokens. A common practice for token pruning is to use attention scores between text and image tokens, assuming higher attention indicates stronger semantic relevance. However, attention scores (Eq. 3) are affected by relative token positions. Prior studies Zhang et al. (2024; 2025b); Luan et al. (2025) show that *text tokens often attend preferentially to spatially proximate visual tokens rather than to semantically relevant ones*. To mitigate this positional bias, we compute attention-like similarity scores between the value vectors ($\mathbf{V}$) of text and image tokens, which are unaffected by positional embeddings.

**Stage 1: Semantic Seed Identification.** Let $\mathbf{V}_{\text{text}} \in \mathbb{R}^{L \times D}$ and $\mathbf{V}_{\text{img}} \in \mathbb{R}^{N \times D}$ denote the value vectors for $L$ text tokens and $N$ visual tokens, respectively, with hidden dimension $D$. We first identify a small set of "semantic seeds"—visual tokens that are strongly aligned with the semantics of the text prompt. For each visual token, we compute a relevance score by averaging the normalized attention it receives from all text tokens:

$$\mathbf{s} = \text{Mean} \left( \text{Softmax} \left( \frac{\mathbf{V}_{\text{text}} \cdot \mathbf{V}_{\text{img}}^T}{\sqrt{D}}, \ \text{dim} = 1 \right), \ \text{dim} = 0 \right) \in \mathbb{R}^N, \tag{9}$$

where the softmax normalizes each text token's attention distribution over visual tokens, and the mean aggregates these scores across all text tokens. We then select the top $1\%$ scoring visual tokens to form the seed set $\mathcal{I}_{\text{seed}}$, where $k_s = \lceil 0.01 \times N \rceil$ is the seed set size.

**Stage 2: Contextual Foreground Refinement.** Semantic seed tokens may only partially cover large or complex objects. To better capture the entire foreground, we expand the query set to include both all text tokens and the initially selected seeds: $\mathbf{V}_{\text{query}} = \mathbf{V}_{\text{text}} \cup \{\mathbf{v}_j^{\text{img}}\}_{j \in \mathcal{I}_{\text{seed}}}$. Let $\mathbf{V}_{\text{query}} \in \mathbb{R}^{(L+k_s) \times D}$ denote the concatenated value vectors from this expanded query set. The refinement score for each visual token is computed by averaging the normalized attention it receives from all query tokens:

$$\mathbf{f} = \text{Mean} \left( \text{Softmax} \left( \frac{\mathbf{V}_{\text{query}} \cdot \mathbf{V}_{\text{img}}^T}{\sqrt{D}}, \ \text{dim} = 1 \right), \ \text{dim} = 0 \right) \in \mathbb{R}^N. \tag{10}$$

The final foreground token set $\mathcal{I}_{\text{fg}}$ is formed by selecting the top-$k_f$ tokens according to $\mathbf{f}$. This ensures that the retained tokens are supported by both textual semantics and key visual features. With the IVC token set $\mathcal{I}_{\text{ivc}}$, the retained token set is $\mathcal{I}_{\text{selected}} = \mathcal{I}_{\text{ivc}} \cup \mathcal{I}_{\text{fg}}$, as summarized in Algorithm 1.

**Pruning Strategy.** A critical design choice is determining the optimal layer for token pruning. Previous works report that *LVLMs are sensitive to token removal in early layers* (Xing et al., 2025; Ye et al., 2025b). Our experiments, however, indicate that this sensitivity primarily arises from the removal of IVC tokens, rather than the pruning operation itself. As shown in Fig. 1, Tab. 6, and

**Algorithm 1** IVC-Prune

**Require:** $\mathbf{V}_{\text{text}} \in \mathbb{R}^{L \times D}$, $\mathbf{V}_{\text{img}} \in \mathbb{R}^{N \times D}$,
$\quad\quad k_c, \ k_f, \ \{\theta_k\}$

**Ensure:** $\mathcal{I}_{\text{selected}} \subseteq \{1, 2, \ldots, N\}$

**// IVC token selection**

1: $V(m) = \sum_{k=0}^{d/2-1} \cos(m\theta_k),$
$\quad U(m) = \sum_{k=0}^{d/2-1} \sin(m\theta_k) \text{ for } m \in [1, N]$

2: $\mathcal{I}_v \leftarrow \arg\text{TopK}(\{V(m)\}, k_c)$

3: $\mathcal{I}_u \leftarrow \arg\text{TopK}(\{U(m)\}, k_c)$

4: $\mathcal{I}_{\text{ivc}} \leftarrow \mathcal{I}_v \cup \mathcal{I}_u$

**// Semantic seed identification**

5: $\mathbf{A}_{\text{seed}} = \dfrac{\mathbf{V}_{\text{text}} \mathbf{V}_{\text{img}}^T}{\sqrt{D}} \in \mathbb{R}^{L \times N}$

6: $\mathbf{A}_{\text{seed}} \leftarrow \text{Softmax}(\mathbf{A}_{\text{seed}}, \ \dim = 1)$

7: $\mathbf{s} = \text{Mean}(\mathbf{A}_{\text{seed}}, \ \dim = 0)$

8: $\mathcal{I}_{\text{seed}} \leftarrow \arg\text{TopK}(\mathbf{s}, k_s), k_s = \lceil 0.01 \times N \rceil$

**// Foreground refinement**

9: $\mathbf{V}_{\text{query}} \leftarrow [\mathbf{V}_{\text{text}}; \mathbf{V}_{\text{img}}[\mathcal{I}_{\text{seed}}, :]]$

10: $\mathbf{f} = \text{Mean}(\text{Softmax}(\dfrac{\mathbf{V}_{\text{query}} \cdot \mathbf{V}_{\text{img}}^T}{\sqrt{D}}))$

11: $\mathcal{I}_{\text{fg}} \leftarrow \arg\text{TopK}(\mathbf{f}, k_f)$

12: **return** $\mathcal{I}_{\text{selected}} = \mathcal{I}_{\text{fg}} \cup \mathcal{I}_{\text{ivc}}$

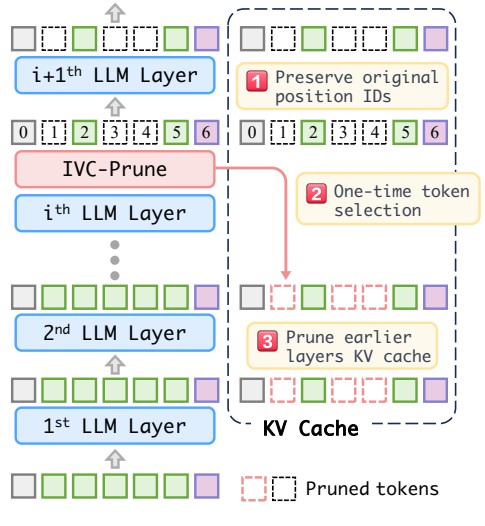

Figure 2: Illustration of the IVC-Prune strategy. Token selection is performed once at layer $i$ on visual tokens, while preserving their original position IDs. The selection decision prunes the corresponding tokens from the KV caches of all earlier layers and is used for subsequent layers.

Tab. 11, performance remains robust even when pruning is applied at all layers, as long as both IVC tokens and foreground are retained. Moreover, consistent with recent findings (Shao et al., 2025; Zhang et al., 2025a), we observe that *attention patterns in intermediate layers are most sensitive to prompt semantics*, whereas shallow and final layers exhibit weaker prompt dependence. Guided by this, as illustrated in Fig. 2, we determine the retained token set once at a selected intermediate layer, while preserving the original position IDs. The selection is then applied to prune the KV caches in all earlier layers, and is also used in subsequent layers. This strategy provides three benefits: (1) minimal overhead from a single pruning operation, (2) prevention of suboptimal selection decisions in shallow layers from adversely affecting subsequent layer computations, and (3) maximized KV cache reduction for enhanced decoding efficiency.

# 4 EXPERIMENTS

## 4.1 EXPERIMENTAL SETTINGS

**Architectures.** We evaluate IVC-Prune on four open-source LVLMs: Qwen2.5-VL (native resolution) Bai et al. (2025), InternVL-2.5 (dynamic resolution) Chen et al. (2024c), DeepSeek-VL2 (dynamic resolution, MoE) Wu et al. (2024), and LLaVA-v1.5 (fixed resolution) Liu et al. (2023).

**Benchmarks.** We evaluate the models across a diverse set of tasks, covering:
*Visual Grounding*: RefCOCO, RefCOCO+ Yu et al. (2016), and RefCOCOg Mao et al. (2016).
*General Reasoning*: SEEDBench (SEED) Li et al. (2023a), MMBench (MMB) Liu et al. (2024a), MMStar (MMS) Chen et al. (2024b), and MME Chaoyou et al. (2023).
*Hallucination Evaluation*: POPE Li et al. (2023c) and HallusionBench (HallB) Guan et al. (2024).
*Real-world Comprehension*: RealWorldQA (RWQA) Corp. (2024).
*OCR*: TextVQA (TVQA) Singh et al. (2019) and AI2D Kembhavi et al. (2016).

**Implementation Details.** We reproduced FastV Chen et al. (2024a), Window FastV (W-FastV) Wen et al. (2025), PDrop Xing et al. (2025), and VScan Zhang et al. (2025a) using the VLMEvalKit framework (Duan et al., 2024) with the default settings. The selection layer $i$ is selected based on empirical performance on a small subset of RefCOCO$_{\text{testA}}$ (or POPE for LLaVA v1.5), and kept fixed for all experiments. Regarding the token preservation ratio, we set $k_c = 10\%$ and $k_f = 40\%$, which

Table 1: Results on visual grounding benchmarks across different LVLMs and token pruning methods. "Average Tokens" is the percentage of visual tokens retained in the KV-cache after pruning. "Rel. Avg." represents the average performance relative to the vanilla. **Bold**: Best. Underline: Second best.

| Models | Method | Average Tokens ↓ | RefCOCO | | | RefCOCO+ | | | RefCOCOg | | Rel. Avg. |
|---|---|---|---|---|---|---|---|---|---|---|---|
| | | | testA | testB | val | testA | testB | val | test | val | |
| **Qwen2.5-VL 7B** | Vanilla | 100% | 92.2 | 84.7 | 89.6 | 88.0 | 74.3 | 82.8 | 86.9 | 86.8 | 100% |
| | FastV | 54% | 74.4 | 76.5 | 75.4 | 68.9 | 66.8 | 67.7 | 75.3 | 74.8 | 84.7% |
| | W-FastV | 54% | 82.9 | 79.8 | 81.8 | 77.4 | 69.0 | 74.0 | 79.2 | 79.5 | 91.0% |
| | PDrop | 61% | 77.6 | 59.1 | 68.7 | 72.1 | 50.1 | 62.6 | 63.7 | 64.4 | 75.4% |
| | VScan | 50% | 90.2 | 82.2 | 86.7 | 84.6 | 70.6 | 79.0 | 83.6 | 83.9 | 96.4% |
| | IVC-Prune | 50% | **92.0** | **84.5** | **89.3** | **87.4** | **74.1** | **82.4** | **86.5** | **86.5** | **99.6%** |
| **InternVL 2.5 8B** | Vanilla | 100% | 94.7 | 86.0 | 90.3 | 91.5 | 78.7 | 85.1 | 87.6 | 87.1 | 100% |
| | FastV | 53% | 87.0 | 77.6 | 81.6 | 82.6 | 70.7 | 76.1 | 77.9 | 78.5 | 90.1% |
| | W-FastV | 53% | 82.9 | 73.6 | 78.8 | 80.1 | 66.6 | 73.5 | 74.4 | 73.4 | 86.0% |
| | PDrop | 56% | 85.0 | 77.8 | 80.8 | 80.9 | 69.9 | 75.1 | 77.7 | 77.0 | 89.0% |
| | IVC-Prune | 50% | **94.2** | **85.7** | **90.3** | **91.1** | **78.2** | **84.8** | **86.9** | **86.4** | **99.5%** |
| **DeepSeek-VL2 Small-16B** | Vanilla | 100% | 96.5 | 92.6 | 95.2 | 94.7 | 87.9 | 91.4 | 93.3 | 93.2 | 100% |
| | FastV | 54% | 94.4 | 89.5 | 92.6 | 91.8 | 83.6 | 87.8 | 90.6 | 90.4 | 96.7% |
| | W-FastV | 54% | 95.0 | 90.4 | 93.6 | 92.5 | 85.2 | 89.2 | 91.3 | 90.9 | 97.8% |
| | PDrop | 57% | 95.7 | 89.1 | 93.0 | 93.7 | 84.5 | 89.0 | 91.5 | 91.3 | 97.7% |
| | IVC-Prune | 52% | **96.0** | **91.8** | **94.5** | **94.0** | **86.6** | **90.3** | **92.4** | **92.2** | **99.0%** |

Table 2: Results on general VQA benchmarks covering reasoning, hallucination, real-world comprehension, and OCR tasks. "A. T." denotes Average Tokens. Green cells surpass the unpruned method.

| Models | Method | A. T.↓ | SEED | MMB | MMS | RWQA | MME | POPE | HallB | TVQA | AI2D | Rel. Avg. |
|---|---|---|---|---|---|---|---|---|---|---|---|---|
| **Qwen2.5-VL 7B** | Vanilla | 100% | 76.7 | 82.4 | 64.2 | 67.8 | 2310.6 | 86.9 | 51.5 | 84.9 | 83.8 | 100% |
| | FastV | 54% | 72.9 | 80.5 | 59.8 | **68.5** | 2242.5 | 86.2 | 54.3 | **84.7** | 81.6 | 98.4% |
| | Win. FastV | 54% | 73.9 | 80.6 | 58.1 | 67.4 | 2235.5 | 85.9 | 49.4 | 83.9 | 81.8 | 96.9% |
| | PDrop | 61% | 74.0 | 78.9 | 57.9 | 66.0 | **2309.7** | 85.7 | 53.3 | 83.9 | 81.2 | 97.5% |
| | VScan | 50% | 74.8 | 80.6 | 59.9 | 68.4 | 2285.0 | 87.3 | **56.5** | 84.3 | 79.1 | 99.1% |
| | IVC-Prune | 50% | **76.7** | **82.6** | 62.9 | 68.2 | 2303.1 | **87.6** | 54.8 | 84.4 | **84.2** | **100.6%** |
| **InternVL 2.5 8B** | Vanilla | 100% | 77.1 | 83.2 | 62.7 | 69.4 | 2344.0 | 89.0 | 50.8 | 79.0 | 84.4 | 100% |
| | FastV | 53% | 74.0 | 81.6 | 62.5 | 65.0 | 2268.3 | 86.7 | 48.8 | 76.8 | 83.1 | 97.0% |
| | Win. FastV | 53% | 73.9 | 82.0 | 57.7 | 65.8 | 2254.0 | 87.1 | 48.3 | 76.3 | 82.8 | 96.1% |
| | PDrop | 56% | 75.4 | 82.7 | 60.5 | 67.7 | **2316.7** | 87.8 | 47.2 | 65.0 | 83.3 | 95.8% |
| | IVC-Prune | 50% | **77.0** | **83.0** | 62.6 | **69.9** | 2308.2 | **88.9** | **50.2** | **78.0** | **84.3** | **99.6%** |
| **DeepSeek-VL2 Small-16B** | Vanilla | 100% | 76.9 | 79.2 | 57.7 | 70.3 | 2128.6 | 89.3 | 43.8 | 83.4 | 82.0 | 100% |
| | FastV | 54% | 75.6 | 78.2 | 55.9 | 69.0 | 2112.8 | 89.2 | 42.7 | 83.1 | 81.0 | 98.6% |
| | Win. FastV | 54% | 76.1 | 78.3 | 56.2 | 68.2 | 2122.4 | 89.0 | 38.1 | 82.3 | 80.5 | 97.3% |
| | PDrop | 57% | 76.8 | 79.1 | 57.3 | 69.7 | **2132.5** | 89.4 | **44.5** | **83.3** | **81.8** | **100.0%** |
| | IVC-Prune | 52% | **77.0** | **79.3** | **57.7** | **70.3** | 2132.2 | **89.5** | 44.3 | 83.0 | **81.8** | **100.1%** |
| **LLaVA-v1.5 7B** | Vanilla | 100% | 64.4 | 60.6 | 34.2 | 54.5 | 1543.1 | 74.5 | 25.8 | 20.7 | 49.1 | 100% |
| | FastV | 30% | 60.1 | 59.8 | 33.5 | 50.2 | 1555.2 | 73.4 | **28.6** | 21.0 | 49.0 | 99.3% |
| | Win. FastV | 30% | 62.2 | 60.2 | 34.1 | 51.6 | **1643.3** | 78.2 | 27.4 | 19.8 | 48.8 | 100.3% |
| | PDrop | 47% | 63.6 | 60.2 | 33.6 | 53.7 | 1600.4 | **79.5** | 28.0 | 18.9 | **49.4** | 100.6% |
| | VScan | 30% | 63.9 | 60.5 | 32.6 | 51.8 | 1637.5 | 78.8 | 28.0 | 21.0 | 49.0 | 101.2% |
| | IVC-Prune | 28% | **64.4** | **60.6** | **34.5** | **54.5** | 1554.4 | 77.6 | 26.7 | **21.1** | 49.2 | **101.3%** |

results in an average token preservation rate of approximately 50%. For LLaVA-v1.5-7B, we use $k_c = 5\%$ and $k_f = 25\%$. More detailed configurations are provided in the Appendix A.2.

## 4.2 RESULTS AND DISCUSSIONS

**Results on Visual Grounding Tasks.** Visual grounding requires precise object localization and thus serves as a strong benchmark for spatial reasoning in LVLMs. As shown in Tab. 1, IVC-Prune reduces roughly 50% of visual tokens, with only marginal drops of 0.4%, 0.5%, and 1.0% across three distinct LVLMs. In contrast, FastV and PDrop struggle on Qwen2.5-VL ($-15.3\%$ and $-24.6\%$)

Table 3: Analysis of inference efficiency on the Qwen2.5-VL-7B evaluated on the RefCOCO$_{testA}$ benchmark. "KV Cache", "Prefill Time", and "Decode Latency" represent per-sample computational costs. "Total Time" measures the complete benchmark execution time. Lower values ($\downarrow$) are better.

| Models | Method | Average Tokens (%) $\downarrow$ | KV Cache (MB) $\downarrow$ | Prefill Time (ms) $\downarrow$ | Decode Latency (ms/token) $\downarrow$ | Total Time (mm'ss) $\downarrow$ | Accuracy (%) $\uparrow$ |
|---|---|---|---|---|---|---|---|
| **Qwen2.5-VL 7B** | Vanilla | 100% | 26.0 (1.0×) | 408 (1.00×) | 65.3 (1.00×) | 60'17 (1.00×) | 92.2 |
| | FastV | 54% | 16.1 (1.6×) | 297 (1.37×) | 62.7 (1.04×) | 51'51 (1.16×) | 74.4 |
| | PDrop | 61% | 16.4 (1.6×) | 315 (1.30×) | 62.8 (1.04×) | 52'23 (1.15×) | 77.6 |
| | IVC-Prune | 50% | 15.9 (1.6×) | 322 (1.27×) | 60.2 (1.08×) | 47'47 (1.27×) | 92.0 |

Table 4: Ablation study on the impact of IVC tokens, using the Qwen2.5-VL-7B model. "w/ " indicates that extra IVC tokens are added to the visual input. "w/o" indicates removing IVC tokens from the visual input. "RC$_{testA}$" and "RC+$_{testA}$" denote the RefCOCO$_{testA}$ and RefCOCO+$_{testA}$.

| Method | Config. | RC$_{testA}$ | RC+$_{testA}$ | SEED | MMB |
|---|---|---|---|---|---|
| Vanilla | Default | 92.2 | 88.0 | 76.7 | 82.4 |
| | w/o IVC | 84.1 | 79.4 | 76.1 | 82.2 |
| IVC-Prune | Default | 92.0 | 87.4 | 76.7 | 82.6 |
| | w/o IVC | 76.0 | 71.3 | 75.2 | 80.6 |
| FastV | Default | 74.4 | 68.9 | 72.9 | 80.5 |
| | w/ IVC | 82.1 | 76.5 | 74.6 | 80.5 |
| PDrop | Default | 77.6 | 72.1 | 74.0 | 78.9 |
| | w/ IVC | 83.9 | 76.5 | 74.6 | 79.2 |

Table 5: Ablation study of applying our method to Qwen2.5-VL models with different parameter sizes (3B, 7B, and 32B).

| Models | Method | RC$_{testA}$ | RC+$_{testA}$ | SEED | MMB |
|---|---|---|---|---|---|
| 3B | Vanilla | 89.6 | 82.5 | 73.8 | 76.7 |
| | FastV | 81.2 | 71.0 | 70.3 | 73.8 |
| | PDrop | 67.6 | 56.8 | 68.4 | 71.7 |
| | IVC-Prune | **89.1** | **81.7** | **73.5** | **75.9** |
| 7B | Vanilla | 92.2 | 88.0 | 76.7 | 82.4 |
| | FastV | 74.4 | 68.9 | 72.9 | 80.5 |
| | PDrop | 77.6 | 72.1 | 74.0 | 78.9 |
| | IVC-Prune | **92.0** | **87.4** | **76.7** | **82.6** |
| 32B | Vanilla | 91.3 | 86.7 | 76.9 | 86.8 |
| | FastV | 74.3 | 67.1 | 70.8 | 81.3 |
| | PDrop | 49.8 | 43.6 | 66.0 | 68.0 |
| | IVC-Prune | **91.1** | **86.3** | **76.7** | **85.8** |

and InternVL 2.5 ($-9.9\%$ and $-11.0\%$), while performing comparatively better on DeepSeek-VL2 ($-3.3\%$ and $-2.3\%$). These results highlight the robustness of IVC-Prune in preserving spatial reasoning performance under aggressive token reduction.

**Results on General VQA Benchmarks.** We evaluate our method on nine diverse VQA benchmarks across four representative LVLMs (Tab. 2). While the average number of retained tokens is comparable to state-of-the-art methods (FastV and PDrop), our method consistently achieves higher performance across all models. Notably, IVC-Prune matches or even surpasses the unpruned vanilla models, achieving average relative scores of $100.6\%$, $99.6\%$, $100.1\%$, and $101.3\%$ for the four LVLMs. In contrast, FastV suffers substantial drops on Qwen2.5-VL ($98.4\%$) and InternVL 2.5 ($97.0\%$), while PDrop degrades on InternVL 2.5 ($95.8\%$). Further experiments on spatial reasoning, video understanding, and additional VQA benchmarks (Appendix A.3) confirm these trends. The results show the robustness of our approach across diverse LVLM architectures and benchmarks.

## 4.3 EFFICIENCY ANALYSIS

Tab. 3 presents an efficiency comparison among vanilla, FastV, and IVC-Prune on Qwen2.5-VL-7B with 8×A100 (40 GB) GPUs. Both FastV and IVC-Prune reduce average token count to roughly half of the baseline. Our prefill time is slightly higher than that of FastV (322ms vs. 297 ms), which is expected given our pruning strategy: we perform token selection at an intermediate layer rather than the shallowest layers. This design retains more semantically relevant tokens, but also increases computation during the prefill stage. However, IVC-Prune applies the pruned token set uniformly across all layers, yielding a further reduction in KV-cache and lower decoding latency (60.2 ms/token vs. 62.7 ms/token for FastV). Total Time measures the complete wall-clock runtime of the benchmark, including both forward computation and token generation, where LVLMs often spend notable time in beam search or sampling operations. Under this realistic measurement, IVC-Prune achieves the shortest total runtime (47'47), demonstrating that the decoding latency reduction effectively compensates for the modest prefill overhead and delivers an improved trade-off between accuracy preservation and practical efficiency.

Table 6: Ablation study comparing IVC tokens with alternative visual token patterns on Qwen2.5-VL-7B. Only the highlighted tokens, pink tokens (pattern-specific) and blue tokens (foreground), are retained as inputs. "Random" selects 15% tokens at random. "C Points" are the corners plus the image center. $\text{IVC}^{5\%-20\%}$ denotes different retain ratios.

| Pattern | None | Random | C Points | Window | Diagonal | $\text{IVC}^{5\%}$ | $\text{IVC}^{10\%}$ | $\text{IVC}^{20\%}$ | Baseline |
|---|---|---|---|---|---|---|---|---|---|
| $\text{RC}_{\text{testA}}$ | 58.0 | 79.3 | 73.5 | 89.0 | 89.8 | 89.1 | 92.8 | **93.3** | 92.2 |
| $\text{RC+}_{\text{testA}}$ | 56.8 | 77.6 | 71.8 | 86.3 | 87.0 | 87.0 | 90.0 | **90.4** | 88.0 |
| $\text{GQA}_{\text{CA}}$ | 90.3 | 92.0 | 91.3 | 92.3 | 92.3 | 93.3 | 93.3 | **93.7** | **93.7** |

Table 7: Ablation of foreground token selection on InternVL-2.5-8B, **with IVC tokens** included in all settings. Stage 1 denotes Semantic Seed Identification. Stage 2 denotes Contextual Foreground Refinement. The variant "Stage 1+2 w/ Text–Image Attention" replaces the value-similarity scoring in Stage 1+2 with conventional text–image attention scores.

| Method | Avg. Tokens | RefCOCO$_{\text{testA}}$ | RefCOCO+$_{\text{testA}}$ | TVQA | MMB |
|---|---|---|---|---|---|
| Vanilla (No Pruning) | 100% | 94.7 | 91.5 | 79.0 | 83.2 |
| Stage 1 | 50% | 93.9 | 90.9 | 76.1 | 83.0 |
| Stage 1 + Stage 2 | 50% | 94.2 | 91.1 | 78.0 | 83.0 |
| Stage 1+2 w/ Text–Image Attention | 50% | 82.2 | 79.3 | 75.7 | 83.1 |

## 4.4 ABLATION STUDIES

**Impact of IVC Tokens.** We analyze our proposed IVC tokens in Tab. 4. Removing IVC tokens causes substantial performance drops on visual grounding tasks on RefCOCO$_{\text{testA}}$: Vanilla degrades from 92.2 to 84.1 (-8.1), and IVC-Prune from 92.0 to 76.0 (-16.0). Conversely, adding IVC tokens to existing methods yields significant improvements: FastV increases from 74.4 to 82.1 (+7.7) and PDrop from 77.6 to 83.9 (+6.3). These results confirm that IVC tokens are essential for spatial reasoning and can be seamlessly integrated into existing pruning methods to enhance their spatial reasoning capabilities. Notably, the impact on non-/weakly-spatial tasks (SEEDBench, MMBench) remains minimal, indicating that IVC tokens specifically target spatial reasoning.

**Effectiveness of IVC Tokens Compared to Alternative Patterns.** To validate the effectiveness of IVC tokens, we compare them with several alternative token patterns on visual grounding benchmarks and GQA$_{\text{choose all}}$ (GQA$_{\text{CA}}$) Zhang et al. (2025d) in Tab. 6. These benchmarks provide foreground annotations, enabling direct analysis of token selection effectiveness. The results confirm that background tokens contribute positively to performance, particularly for visual grounding. On RefCOCO, $\text{IVC}^{10\%}$ outperforms the full-token baseline. On the less spatially focused GQA benchmark, all variants achieve comparable accuracy, with $\text{IVC}^{20\%}$ matching the baseline at 93.7%. Thus, we adopt $\text{IVC}^{10\%}$ as our default configuration, delivering optimal performance with high efficiency.

**Ablation of Foreground Token Selection.** Tab. 7 evaluates variations of our foreground selection strategy, where all configurations include IVC tokens by default and differ only in the selection mechanism. Using only the semantic seed identification (stage 1) recovers most of the performance but still underperforms the complete two-stage method, particularly on TextVQA (-1.9). This highlights the importance of the refinement stage for better capturing the complete foreground. Replacing the value-similarity scores with text-image attention scores results in a substantial performance drop in grounding tasks (*e.g.*, -12.0 on RefCOCO$_{\text{testA}}$), suggesting that attention scores are suboptimal, likely due to positional bias. Interestingly, MMBench performance remains unchanged across settings, indicating that this benchmark may be less sensitive to the specifics of visual tokens.

**Effectiveness Across Parameter Scales.** Tab. 5 evaluates our method across different model scales (3B, 7B, and 32B parameters). Across all scales, IVC-Prune consistently preserves the performance of the vanilla model. The robustness of IVC-Prune across diverse scales demonstrates that our approach is not limited by model capacity, supporting its applicability to a wide range of LVLMs.

## 5 DISCUSSION AND LIMITATIONS

### 5.1 DISCUSSION ON NOVELTY AND RELATION TO PRIOR WORK

While token pruning is an established field, IVC-Prune introduces distinct theoretical and method-ological innovations compared to prior token pruning approaches:

**Theoretical analysis for spatial reasoning in LVLMs.** We present the first theoretical characteri-zation of the mathematical structure of RoPE within LVLMs, revealing that certain token positions function as Implicit Visual Coordinate (IVC) tokens. These tokens encode absolute spatial infor-mation essential for object localization at arbitrary image resolutions. This analysis offers novel mechanistic insights into how LVLMs localize objects at arbitrary resolutions, which is a fundamental property previously unexplored in pruning literature.

**Explanation of early-layer pruning sensitivity.** Prior works (Xing et al., 2025; Ye et al., 2025b) reported severe performance drops when pruning early transformer layers, without clarifying the cause. Our controlled ablations (Tab. 6, Tab. 11) show that sensitivity arises from the removal of IVC tokens rather than pruning itself.

**Safe early-layer KV-cache pruning.** Guided by the IVC analysis, we implement a single-selection strategy that enables pruning of early-layer KV caches while preserving performance. This achieves maximal cache reduction and faster decoding, whereas prior training-free methods avoid early-layer pruning entirely due to quality loss.

**Cross-Architecture Robustness.** We introduce a two-stage semantic foreground token selection method that leverages value-vector similarity, followed by semantic seeding and contextual refine-ment. This design effectively reduces positional bias from attention distributions. Our method generalizes across diverse image inputs, yielding competitive results on four LVLM families and twenty benchmarks, demonstrating that it is architecture-agnostic.

### 5.2 LIMITATIONS

Although IVC-Prune performs well across diverse LVLMs and tasks, several limitations remain. First, our fixed pruning ratios, while effective in both image and video settings, are not dynamically adapted to task-specific visual or temporal complexity, which may lead to suboptimal pruning in certain scenarios. Second, the pruning layer is selected based on validation performance on a small subset of the benchmark. While it is effective in practice, this underscores the need for automated strategies to identify the optimal pruning layer and motivates further interpretability studies into the distinct functional roles of different layers in LVLMs. Finally, the identification of IVC tokens is inherently tied to the mathematical structure of RoPE. Extending this framework to architectures using alternative positional encodings (*e.g.* learned 2D embeddings) will require additional theoretical and empirical validation.

## 6 CONCLUSION

In this work, we present a new insight into how LVLMs perform spatial reasoning: LVLMs inherently establish an implicit visual coordinate system through the mathematical structure of RoPE, using specific token positions as spatial coordinates. Based on this, we introduce IVC-Prune, a novel training-free pruning method that preserves both crucial Implicit Visual Coordinate (IVC) tokens and semantically aligned foreground tokens. Experiments across diverse LVLM architectures and benchmarks demonstrate that IVC-Prune reduces computational costs (*e.g.*, 50% KV-cache reduction) with negligible performance loss, and in some cases even surpasses the unpruned vanilla method. In the future, we plan to extend IVC-Prune with dynamic, task-adaptive pruning ratios and explore its integration into training-time architecture design. We hope our findings will motivate further research into positional encoding mechanisms and their role in spatial reasoning within LVLMs.

## ACKNOWLEDGMENTS

This work was supported partially by the National Key Research and Development Program of China (2023YFC2705700), NSFC 62222112, 62176186, and the NSF of Hubei Province of China (2024AFB245).

## REPRODUCIBILITY

We provide comprehensive details to ensure full reproducibility of our experimental results in Section 4.1, Appendix A.1 and Appendix A.2. All evaluations are conducted using the default inference settings on 8×A100 40 GB GPUs (8×H800 80 GB for the 32B and video experiments). The code and experimental configurations are publicly available at https://github.com/FireRedTeam/IVC-Prune.

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

# A  APPENDIX

## A.1  DETAILS OF EVALUATION BENCHMARKS

**Visual Grounding:** RefCOCO and RefCOCO+ Yu et al. (2016) (testA, testB, val), and Ref-COCOg Mao et al. (2016) (test, val).

**General Reasoning:** SEEDBench$_{image}$ (SEED) Li et al. (2023a), MMBench$_{DEV\_EN\_V11}$ (MMB) Liu et al. (2024a), AI2D$_{test}$ (AI2D) Kembhavi et al. (2016), MMStar (MMS) Chen et al. (2024b), MME Chaoyou et al. (2023), MMBench$_{DEV\_EN}$, MMBench$_{DEV\_CN\_V11}$, MMBench$_{DEV\_CN}$ Liu et al. (2024a).

**Hallucination Evaluation**: POPE Li et al. (2023c), HallusionBench Guan et al. (2024).

**Real-world Comprehension**: RealWorldQA (RWQA) Corp. (2024), A-OKVQA Schwenk et al. (2022).

**OCR**: TextVQA (TVQA) Singh et al. (2019) and AI2D Kembhavi et al. (2016).

**Science Knowledge:** ScienceQA (SQA) Lu et al. (2022).

**Spatial Reasoning:** SpatialEval Wang et al. (2024).

**Video Understanding:** MVBench Li et al. (2024a), Video-MME Fu et al. (2025), and MLVU Zhou et al. (2025).

For all benchmarks, we follow the standardized evaluation protocol adopted in VLMEvalKit Duan et al. (2024), employing GPT-4.1 as the judge model for question-answer scoring. For GQA$_{choose\ all}$ Zhang et al. (2025d), we report performance on the *ChooseAttr*, *ChooseCat*, and *ChooseRel* subsets.

## A.2  DETAILED CONFIGURATION OF DIFFERENT LVLMS

Different LVLMs adopt distinct image preprocessing pipelines and architecture depths, as illustrated in Fig. 3. Consequently, the application of IVC-Prune requires minor model-specific adjustments. To ensure reproducibility, this section provides the detailed configuration for each model.

**Layer Selection Protocol.**  The pruning layer $i$ is determined based on validation performance on a small subset of RefCOCO$_{testA}$ (or POPE for LLaVA v1.5). Once the optimal layer is determined for each model, the same layer configuration is consistently applied across all benchmarks and tasks to ensure fair comparison and reproducibility.

**Model-specific Settings.**

- **Qwen2.5-VL:** We apply IVC-Prune at layer 16 in the 7B model (28 layers total), layer 22 in the 3B model (32 layers total), and layer 35 in the 32B model (64 layers total).
- **InternVL 2.5:** We perform IVC-Prune at layer 16 (32 layers total). The pruning is first applied to the thumbnail image. Then, the identified foreground tokens are mapped to the corresponding positions in tiled images. IVC tokens are computed independently within each tiled image and the thumbnail image.
- **DeepSeek-VL2:** We apply IVC-Prune at layer 17 (27 layers total). Similar to InternVL 2.5, we first perform pruning on the thumbnail (excluding padding areas) and map foreground tokens to tiled images. Since DeepSeek-VL2 uses special tokens to separate lines in tiles, we select IVC tokens within each line while preserving all special tokens. We avoid pruning layer 0 as DeepSeek-VL2 relies on the layer 0 KV-cache length to manage the prefilling stage.
- **LLaVA v1.5:** We perform IVC-Prune at layer 15 (32 layers total). Layer 0 is excluded from pruning as removing its KV-cache leads to significant performance degradation Chen et al. (2024a).

**Baseline Method Settings.**  We reproduce FastV Chen et al. (2024a) and PDrop Xing et al. (2025) with the following configurations:

- **FastV:** $K = 2, R = 50\%$ for Qwen2.5-VL, InternVL 2.5, and DeepSeek-VL2; $K = 2, R = 75\%$ for LLaVA v1.5. We use the KV-cache compatible implementation.
- **PDrop:** $K \in \{8, 16, 24\}$ with $\lambda \in \{0.7, 0.5, 0.5\}$ for Qwen2.5-VL, InternVL 2.5, and DeepSeek-VL2; $\lambda \in \{0.5, 0.5, 0.5\}$ for LLaVA v1.5.

**Video Benchmark Implementation.** For experiments on video datasets, we adopt a consistent per-frame pruning strategy. IVC-Prune is applied to each video frame independently, retaining 50% of the original visual tokens for every frame.

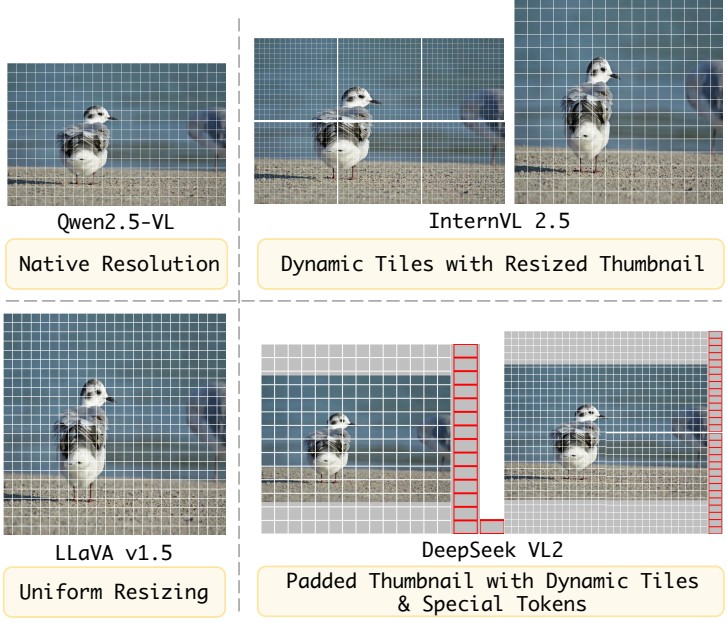

Figure 3: Comparison of image preprocessing strategies in different LVLMs. White squares represent image tokens. Red squares indicate special tokens introduced in DeepSeek-VL2.

## A.3 FURTHER EXPERIMENTS

**Analysis of Spatial Reasoning Benchmark Results.** Tab. 8 demonstrates the effectiveness of our method on spatial reasoning tasks. Across all evaluated LVLMs, our IVC-Prune achieves the highest *Overall* scores compared with other pruning methods.

**Analysis of Video Understanding Benchmark Results.** Tab. 9 demonstrates the effectiveness of our method on video understanding tasks. By retaining only 50% of the original visual tokens, our approach consistently meets or exceeds the performance of the baseline, achieving average relative scores of 100.1% for Qwen2.5-VL-7B and 100.3% for InternVL 2.5-8B. These results underscore that our pruning strategy is robust and effective for processing temporal data.

**Analysis of VQA Benchmark Results.** The results in Tab. 10 further corroborate the effectiveness of our proposed method. Across four distinct Large Vision-Language Models (LVLMs), our approach consistently demonstrates a superior efficiency-performance trade-off. While retaining only about 50% of the visual tokens for most models (and as low as 28% for LLaVA-v1.5), our method achieves average relative performance scores of 100.2%, 100.0%, 100.1%, and 101.3%. These scores not only meet or exceed the baseline but also consistently surpass competing pruning methods like FastV and PDrop, often while using fewer tokens. This underscores the robustness of our strategy in preserving essential visual information for complex VQA tasks across diverse model architectures.

## A.4 EXTENDED RESULTS AND ANALYSIS FOR FIG. 1

Tab. 11 presents detailed results of visual grounding performance across multiple LVLMs and input configurations, extending the summary in Fig. 1. For most models (Qwen2.5-VL-7B/3B and

Table 8: Evaluation results on spatial reasoning benchmark.

| Models | Method | Average Tokens ↓ | SpatialEval | | | | |
| --- | --- | --- | --- | --- | --- | --- | --- |
| | | | Mazenav | Spatialgrid | Spatialmap | Spatialreal | *Overall* |
| Qwen2.5-VL 7B | Vanilla | 100% | 32.8 | 84.1 | 68.3 | 70.4 | 62.0 |
| | FastV | 54% | 33.5 | 78.3 | 68.1 | 69.6 | 60.2 |
| | PDrop | 61% | 26.8 | 75.5 | 66.8 | 63.7 | 56.6 |
| | IVC-Prune | 50% | **35.2** | **83.4** | **69.9** | **70.4** | **63.0** |
| InternVL 2.5 8B | Vanilla | 100% | 33.5 | 76.2 | 63.5 | 65.9 | 58.0 |
| | FastV | 53% | 28.1 | 54.5 | 52.6 | 60.0 | 45.5 |
| | PDrop | 56% | **32.8** | 66.1 | 60.8 | 61.5 | 53.5 |
| | IVC-Prune | 50% | 32.7 | **76.1** | **63.8** | **68.1** | **57.8** |
| DeepSeek-VL2 Small-16B | Vanilla | 100% | 32.1 | 75.2 | 59.3 | 60.0 | 55.6 |
| | FastV | 54% | 32.6 | 65.9 | 58.0 | 58.5 | 52.3 |
| | PDrop | 57% | **33.5** | 73.8 | 59.1 | **61.5** | 55.6 |
| | IVC-Prune | 52% | 32.3 | **75.3** | **59.3** | 59.3 | **55.7** |
| LLaVA-v1.5 7B | Vanilla | 100% | 31.3 | 30.1 | 43.5 | 38.5 | 35.0 |
| | FastV | 30% | 30.4 | 29.5 | **44.1** | 35.5 | 34.5 |
| | PDrop | 47% | **30.8** | 29.1 | 44.0 | 35.5 | 34.5 |
| | IVC-Prune | 28% | 30.3 | **30.0** | 43.1 | **38.5** | **34.6** |

Table 9: Evaluation results on video understanding benchmark.

| Models | Method | Average Tokens ↓ | MVBench 64 frame | VideoMME | | | | MLVU | | Rel. Avg. |
| --- | --- | --- | --- | --- | --- | --- | --- | --- | --- | --- |
| | | | | w/o subs | short | medium | long | M-Aug | G-Aug | |
| Qwen2.5-VL 7B | Vanilla | 100% | 67.7 | 63.1 | 74.0 | 62.9 | 52.6 | 65.1 | 5.65 | 100% |
| | FastV | 54% | **67.7** | 62.7 | 74.1 | 62.7 | 51.3 | 65.0 | 5.52 | 99.2% |
| | PDrop | 61% | **67.7** | 63.1 | 74.0 | 62.9 | 52.6 | 64.0 | 5.53 | 99.5% |
| | IVC-Prune | 50% | **67.7** | **63.4** | **74.3** | **63.1** | **52.7** | **65.7** | **5.55** | **100.1%** |
| InternVL 2.5 8B | Vanilla | 100% | 70.4 | 63.6 | 75.8 | 62.8 | 52.3 | 68.6 | 4.79 | 100% |
| | FastV | 53% | 69.2 | 62.9 | 73.3 | 61.6 | **53.7** | 65.7 | **4.77** | 98.6% |
| | PDrop | 56% | 68.6 | 62.1 | 73.4 | 62.8 | 50.1 | 66.5 | 4.65 | 97.4% |
| | IVC-Prune | 50% | **70.1** | **64.3** | **76.4** | **63.2** | 53.1 | **68.3** | 4.74 | **100.3%** |

DeepSeek-VL2-Small-16B), restricting the inputs to only foreground tokens leads to a substantial performance drop across all benchmarks. Remarkably, augmenting these foreground tokens with just 10% IVC tokens not only recovers performance but often surpasses the unpruned vanilla baseline. This suggests that IVC tokens provide effective visual coordinates required for accurate object localization.

An exception arises with InternVL 2.5-8B, where the foreground-only setting retains comparatively high accuracy. We hypothesize that this robustness stems from its distinctive image processing strategy, which uses fixed-size thumbnails and tiled images with pre-defined aspect ratios. This fixed-size input may implicitly encode positional and boundary information, reducing the model's reliance on implicit visual coordinates. In contrast, the other LVLMs operate on variable-resolution inputs and thus appear more sensitive to the absence of IVC tokens. Nevertheless, even for InternVL 2.5, adding IVC tokens yields further gains. This confirms the universal value of IVC tokens.

## A.5 ANALYSIS OF CLUSTERING- OR MERGING- BASED TOKEN REDUCTION METHODS

In this section, we provide a detailed analysis of the limitations inherent in clustering or merging-based token reduction mechanisms (*e.g.*, Llava-PruMerge Shang et al. (2024), SparseVLM Zhang et al. (2025c), and PACT Dhouib et al. (2025)), specifically concerning their impact on spatial reasoning capabilities.

In prior work on token pruning, Position IDs are typically handled in one of two ways: preserving the original position IDs of retained tokens or reassigning position IDs to produce a contiguous index sequence in the pruned representation. Retaining the original position IDs maintains spatial

Table 10: Comprehensive evaluation results on additional VQA benchmarks.

| Models | Method | A. T.↓ | A-OKVQA | SQA$_{TEST}$ | MMB$_{EN}$ | MMB$_{CN\_V11}$ | MMB$_{CN}$ | Rel. Avg. |
|--------|--------|--------|---------|----------|--------|-----------|--------|-----------|
| **Qwen2.5-VL 7B** | Vanilla | 100% | 86.5 | 88.7 | 83.4 | 81.4 | 82.2 | 100% |
| | FastV | 54% | 86.4 | 85.8 | 81.8 | 80.0 | 80.8 | 98.2% |
| | PDrop | 61% | 85.6 | **87.5** | 80.8 | 79.5 | 80.5 | 98.0% |
| | IVC-Prune | 50% | **86.7** | 87.0 | **83.8** | **82.6** | **82.7** | **100.2%** |
| **InternVL 2.5 8B** | Vanilla | 100% | 87.2 | 98.1 | 83.9 | 82.9 | 83.1 | 100% |
| | FastV | 53% | 86.6 | 97.4 | 82.5 | 80.8 | 81.9 | 98.6% |
| | PDrop | 56% | 86.9 | 97.9 | 83.4 | 81.5 | 81.9 | 99.1% |
| | IVC-Prune | 50% | **87.2** | **98.2** | **83.9** | **83.0** | 82.9 | **100.0%** |
| **DeepSeek-VL2 Small-16B** | Vanilla | 100% | 86.9 | 96.9 | 80.8 | 78.8 | 79.7 | 100% |
| | FastV | 54% | 85.9 | 96.3 | 79.7 | 77.9 | 78.5 | 98.8% |
| | PDrop | 57% | **86.7** | **96.8** | **80.8** | 78.9 | 79.8 | 100.0% |
| | IVC-Prune | 52% | 86.6 | **96.8** | **80.8** | **79.0** | **80.1** | **100.1%** |
| **LLaVA-v1.5 7B** | Vanilla | 100% | 78.9 | 66.4 | 63.3 | 41.3 | 41.8 | 100% |
| | FastV | 30% | 79.0 | 66.3 | 62.5 | 40.9 | 42.2 | 99.7% |
| | PDrop | 47% | **79.2** | 66.3 | 62.8 | 40.9 | 42.1 | 99.8% |
| | IVC-Prune | 28% | 78.8 | **66.4** | **63.3** | **42.8** | **43.0** | **101.3%** |

Table 11: Extended and detailed results corresponding to Fig. 1: Performance comparison on visual grounding benchmarks across different LVLMs under various input settings.

| Inputs | Method | RefCOCO | | | RefCOCO+ | | | RefCOCOg | |
|--------|--------|-------|-------|------|-------|-------|------|------|------|
| | | testA | testB | val | testA | testB | val | test | val |
| **Qwen2.5-VL 7B** | Vanilla | 92.2 | **84.7** | 89.6 | 88.0 | 74.3 | 82.8 | **86.9** | 86.8 |
| | Foreground tokens | 58.0 | 39.4 | 49.0 | 56.8 | 39.4 | 48.6 | 44.6 | 44.9 |
| | Foreground + 10% IVC tokens | **92.8** | 82.9 | **89.8** | **89.9** | **77.2** | **86.2** | 86.0 | **87.1** |
| **Qwen2.5-VL 3B** | Vanilla | 89.6 | 83.4 | 87.6 | 82.5 | 71.4 | 77.9 | 84.3 | 83.9 |
| | Foreground tokens | 61.9 | 51.0 | 55.1 | 57.2 | 48.4 | 51.6 | 52.2 | 53.6 |
| | Foreground + 10% IVC tokens | **92.9** | **86.7** | **90.9** | **88.9** | **82.0** | **86.2** | **89.5** | **90.0** |
| **InternVL 2.5 8B** | Vanilla | **94.7** | 86.0 | 90.3 | 91.5 | 78.7 | 85.1 | 87.6 | 87.1 |
| | Foreground tokens | 92.4 | 86.9 | 89.2 | 91.3 | 84.9 | 88.3 | 88.5 | 87.6 |
| | Foreground + 10% IVC tokens | 93.8 | **89.3** | **91.1** | **92.8** | **85.8** | **89.5** | **90.8** | **89.6** |
| **DeepSeek-VL2 Small-16B** | Vanilla | **96.5** | 92.6 | **95.2** | 94.7 | 87.9 | 91.4 | 93.3 | 93.2 |
| | Foreground tokens | 21.6 | 18.4 | 20.2 | 19.6 | 17.3 | 18.8 | 17.4 | 17.2 |
| | Foreground + 10% IVC tokens | 96.0 | **92.9** | 94.7 | **94.9** | **88.5** | **91.6** | **93.8** | **93.8** |

consistency, whereas clustering or merging approaches, as in SparseVLM, generate aggregated tokens that require position ID reconstruction. These reconstructed indices do not correspond to precise coordinates in the original visual grid, resulting in a loss of spatial location fidelity.

To validate this observation, we reproduced SparseVLM using Qwen2.5-VL 7B as the backbone and compared two configurations: the default implementation, which includes the clustering and merging step, and a modified variant in which clustering is disabled. Results, shown in Tab. 12, reveal that the default configuration suffers severe degradation on the RefCOCO spatial grounding benchmarks, achieving only **15.3%** of the baseline accuracy. In contrast, the variant without clustering recovers performance to **76.1%**, confirming that the merging operation and consequent position ID reconstruction are the dominant sources of error.

Interestingly, performance on general VQA tasks remains comparable to the baseline when clustering is used, indicating that semantic information is largely preserved while spatial structure is compromised. These findings are consistent with prior work Chien et al. (2025) and lead to an important conclusion: preserving the original position IDs is essential for tasks involving fine-grained spatial reasoning.

Table 12: Ablation study on the effect of clustering in SparseVLM.

| Method | A. T. | RefCOCO | | | RefCOCO+ | | | RefCOCOg | | Rel. Avg. |
|---|---|---|---|---|---|---|---|---|---|---|
| | | testA | testB | val | testA | testB | val | test | val | |
| Vanilla | 100% | 92.2 | 84.7 | 89.6 | 88.0 | 74.3 | 82.8 | 86.9 | 86.8 | 100% |
| IVC-Prune | 50% | 92.0 | 84.5 | 89.3 | 87.4 | 74.1 | 82.4 | 86.5 | 86.5 | 99.6% |
| SparseVLM w/o clustering | 49% | 77.2 | 61.2 | 69.1 | 71.7 | 53.5 | 63.2 | 63.2 | 63.6 | 76.1% |
| SparseVLM w clustering | 51% | 14.1 | 13.8 | 14.2 | 12.6 | 11.7 | 12.5 | 12.3 | 13.6 | 15.3% |

| Method | A. T. | SEED | MMB | MMS | RWQA | MME | POPE | HallB | TVQA | AI2D | Rel. Avg. |
|---|---|---|---|---|---|---|---|---|---|---|---|
| Vanilla | 100% | 76.7 | 82.4 | 64.2 | 67.8 | 2310.6 | 86.9 | 51.5 | 84.9 | 83.8 | 100% |
| IVC-Prune | 50% | 76.7 | 82.6 | 62.9 | 68.2 | 2303.1 | 87.6 | 54.8 | 84.4 | 84.2 | 100.6% |
| SparseVLM w/o clustering | 49% | 74.9 | 79.6 | 45.6 | 61.4 | 2279.6 | 85.9 | 53.5 | 83.7 | 82.3 | 94.9% |
| SparseVLM w clustering | 51% | 74.8 | 82.0 | 61.3 | 67.8 | 2320.0 | 86.8 | 54.5 | 84.5 | 82.5 | 99.6% |

## A.6 ANALYSIS OF HIGH-RESOLUTION OCR BENCHMARKS.

We evaluate the proposed IVC-Prune on three high-resolution OCR benchmarks DocVQA Mathew et al. (2021), InfoVQA Mathew et al. (2022), and OCRBench Liu et al. (2024b) using the Qwen2.5-VL 7B model. Results are reported in Tab. 13. On DocVQA and InfoVQA, IVC-Prune achieves accuracy comparable to the Vanilla model, and consistently outperforms FastV and PDrop, suggesting that the proposed method is well-suited for high-resolution VQA scenarios.

Table 13: Evaluation results on high-resolution OCR benchmark.

| Method | Avg. Tokens | DocVQA | InfoVQA | OCRBench |
|---|---|---|---|---|
| Vanilla | 100% | 94.9 | 81.7 | 88.4 |
| FastV | 52% | 93.9 | 76.4 | **67.3** |
| PDrop | 52% | 93.8 | 69.4 | 62.0 |
| IVC-Prune | 50% | **94.4** | **80.9** | 66.3 |

In contrast, all pruning methods suffer substantial degradation on OCRBench. To understand this discrepancy, we conduct a fine-grained analysis across OCRBench's task categories (Tab. 14). The most pronounced drops occur in **recognition-focused** tasks, specifically Text Recognition and Handwritten Mathematical Expression Recognition. Meanwhile, VQA categories remain largely unaffected.

We attribute this gap to the nature of recognition-focused tasks: images typically contain densely packed characters or symbols with minimal background. Accurate recognition relies on preserving fine-grained visual details. Under a uniform 50% pruning ratio, a substantial portion of tokens encoding these details are removed, leading to inevitable performance loss.

Visualization examples in Tab. 15 further support this analysis. Failure cases originate from recognition-focused tasks and relatively low-resolution images. These inputs contain mostly foreground tokens relevant to text or symbolic content, leaving little redundant information to prune. In contrast, VQA-oriented inputs retain sufficient visual context even after pruning, sustaining strong performance.

Overall, our findings highlight that the performance drop on OCRBench is not a general failure of the proposed IVC-Prune for high-resolution inputs, but rather a limitation in handling high-density text recognition. This suggests that *adaptive* or *task-aware* pruning strategies, which account for token density and semantic importance, may be necessary to maintain accuracy in such scenarios.

## A.7 ADDITIONAL RESULTS UNDER EXTREMELY LOW TOKEN BUDGETS

Following the evaluation protocol in PDrop, we performed experiments on LLaVA-v1.5-7B to assess the behavior of IVC-Prune under extremely low token retention rates. Specifically, we evaluated three settings corresponding to 33.3%, 22.2%, and 11.1% of the original token budget. As shown in Tab. 16, IVC-Prune exhibits remarkable robustness under these aggressive token reduction scenarios.

Table 14: Task-level breakdown of OCRBench performance. Parentheses indicate the relative performance compared to Vanilla.

| Task Category | Vanilla | IVC-Prune | FastV | PDrop |
|---|---|---|---|---|
| Scene Text-centric VQA | 17.9 | 17.6 (98.3%) | 17.7 (98.9%) | 17.7 (98.9%) |
| Doc-oriented VQA | 18.0 | 17.1 (95.0%) | 16.7 (92.8%) | 14.7 (81.7%) |
| Key Information Extraction | 18.2 | 16.8 (92.3%) | 13.0 (71.4%) | 11.8 (64.8%) |
| Text Recognition | 26.9 | 14.0 (52.0%) | 18.3 (68.0%) | 17.6 (65.4%) |
| Handwritten Mathematical Expression Recognition | 7.4 | 0.8 (10.8%) | 1.6 (21.6%) | 0.2 (2.7%) |
| **Final Score** | 88.4 | 66.3 (75.0%) | 67.3 (76.1%) | 62.0 (70.1%) |

Table 15: Qualitative visualization of IVC-Prune performance on the OCRBench benchmark. Gray regions indicate tokens pruned by the model. For high-resolution images, red bounding boxes were added post hoc to highlight locations relevant to the ground-truth answer.

| Reserved Tokens | Question | Resolution | Answer | Prediction | Correct |
|---|---|---|---|---|---|
|  | what is written in the image? | 196×56 | rcaihttnu | The text in the image reads "rcaintnu". | ✗ |
|  | what is written in the image? | 112×28 | CENTRE | century | ✗ |
|  | what is the number in the image? | 196×28 | 100972 | The image contains the handwritten text "$\lambda 0 = 72$". | ✗ |
|  | What is the period of registry observation taken into consideration for ' bilateral arthroplasty ' ? | 1652x2352 | 14 years | The period of registry observation considered for 'bilateral arthroplasty' is 14 years. | ✓ |
|  | What is total percentage of businesses laying of staff and reducing staff hours? | 1148x1876 | 68.3% | To find the total percentage of businesses that either laid off staff or reduced staff hours, you need to add the percentages for each category: - Percentage of businesses that laid off staff: 28.4% - Percentage of businesses that reduced staff hours or shifts: 39.9% Adding these two percentages together: $28.4\% + 39.9\% = 68.3\%$ So, the total percentage of businesses that either laid off staff or reduced staff hours is 68.3%. | ✓ |

Notably, at 11.1% token retention, IVC-Prune sustains 97.2% of the baseline performance. This robustness highlights the method's capacity to maintain critical tokens.

Table 16: Results under extremely low token budgets on LLaVA-v1.5 7B.

| Method | A. T. ↓ | SEED | MMB | MMS | RWQA | MME | POPE | HallB | TVQA | AI2D | Rel. Avg. |
|---|---|---|---|---|---|---|---|---|---|---|---|
| Vanilla | 576 (100%) | 64.4 | 60.6 | 34.2 | 54.5 | 1543.1 | 74.5 | 25.8 | 20.7 | 49.1 | 100% |
| FastV | 192 (33.3%) | 61.2 | 60.2 | 33.4 | 51.6 | 1572.7 | 74.8 | **29.0** | **21.8** | 48.8 | 100.7% |
| PDrop | 192 (33.3%) | 60.0 | 54.6 | 31.9 | 51.2 | **1607.6** | **80.1** | 25.7 | 17.0 | 48.6 | 95.9% |
| IVC-Prune | 192 (33.3%) | **64.5** | **60.7** | **34.5** | 54.4 | 1567.7 | 76.9 | 26.2 | 21.1 | **49.1** | **101.0%** |
| FastV | 128 (22.2%) | 57.2 | 58.3 | 33.1 | 47.7 | 1462.1 | 67.8 | **28.1** | 18.7 | 48.0 | 94.7% |
| PDrop | 128 (22.2%) | 56.0 | 49.8 | 31.1 | 50.8 | 1565.4 | 77.2 | 24.4 | 14.8 | 45.1 | 90.7% |
| IVC-Prune | 128 (22.2%) | **64.5** | **60.7** | **34.0** | 54.4 | 1525.6 | 77.0 | 26.2 | 20.1 | **49.1** | **100.0%** |
| FastV | 64 (11.1%) | 45.8 | 40.2 | 28.4 | 38.2 | 1109.7 | 31.9 | **27.1** | 6.6 | 45.8 | 70.6% |
| PDrop | 64 (11.1%) | 46.3 | 39.2 | 28.3 | 46.5 | 1205.7 | 51.0 | 23.6 | 8.6 | 46.2 | 75.4% |
| IVC-Prune | 64 (11.1%) | **64.2** | **60.4** | **33.9** | **54.1** | **1511.3** | **75.4** | 24.8 | 16.9 | **49.1** | **97.2%** |

## A.8 ABLATION OF TOKEN ALLOCATION STRATEGY

To determine the optimal balance between spatial structure and semantic content, we conducted an ablation study on the IVC token ratio. Operating under a fixed 50% total token budget, we evaluated IVC allocations of 5%, 10%, and 20% using the Qwen2.5-VL 7B model. The remaining budget in each setting is assigned to foreground semantic tokens.

The results, summarized in Tab. 17, demonstrate that allocating **10%** of tokens to IVC yields the best performance. Lowering the ratio to 5% results in a performance drop due to insufficient spatial fidelity, while increasing it to 20% degrades performance by restricting the budget available for semantic foreground tokens.

Table 17: Ablation study on token allocation strategy under a 50% total token budget. Experiments were conducted on Qwen2.5-VL 7B. The "IVC Token" column denotes the percentage of total tokens.

| IVC Token Ratio | RefCOCO$_{testA}$ | RefCOCO+$_{testA}$ | SeedBench | MMBench |
|---|---|---|---|---|
| 5% | 91.8 | 87.2 | 76.6 | 82.5 |
| **10% (Ours)** | **92.0** | **87.4** | **76.7** | **82.6** |
| 20% | 91.3 | 87.0 | 76.6 | 82.3 |

## A.9 VISUALIZATION OF IVC TOKENS

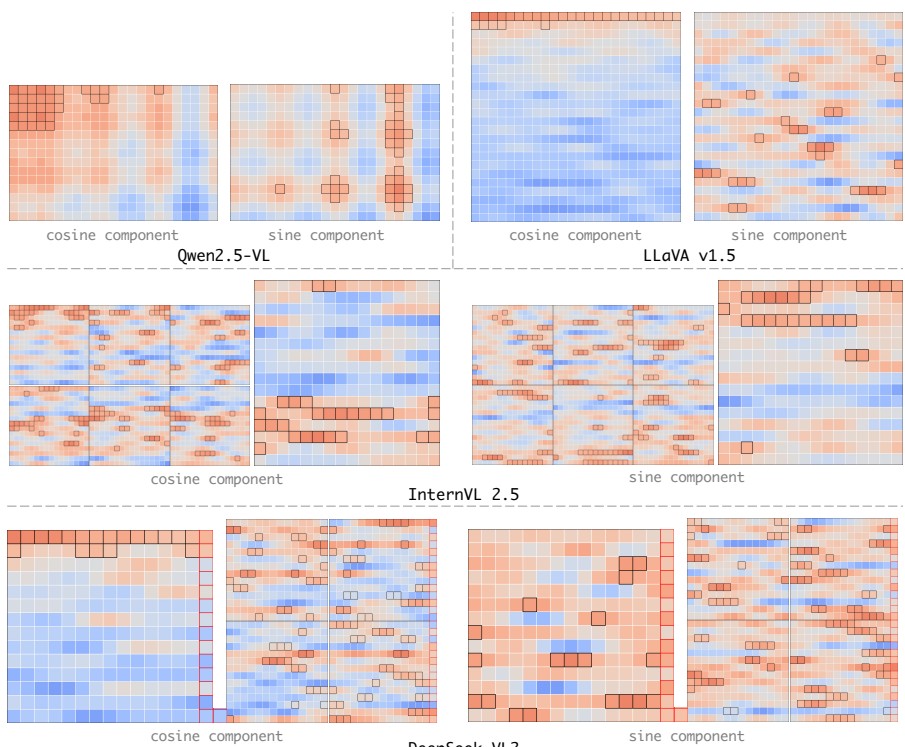

Figure 4: Visualization of the positional embedding scores for four LVLMs, where cosine $(V(m))$ and sine $(U(m))$ components are summed over all dimensions as in Eqs. 5 and 7. Black squares denote the selected 10% IVC tokens, and red squares indicate the special tokens introduced in DeepSeek-VL2. Note that IVC tokens are determined solely by position and are independent of the content.

