# OpenReview forum: "IVC-Prune: Revealing the Implicit Visual Coordinates in LVLMs for Vision Token Pruning"
_ICLR.cc/2026/Conference — ICLR 2026 Poster_

### Official Review · Reviewer_HRNj · 2025-10-27

**Soundness:** 2
**Presentation:** 3
**Contribution:** 3
**Rating:** 6
**Confidence:** 2

**Summary:**

This work presents a novel method for visual token pruning for VLMs. Analyses reveal that there are specific token positions as implicit visual coordinates (IVC tokens) that are essential for spatial reasoning. The proposed method, IVC-prune, is based on this insight and retains both IVC tokens and semantically-related tokens. IVC tokens are identified by theoretically analyzing ROPE, while semantic tokens to keep are identified through a two-stage process of discovery and refinement. Experiments with 4 VLMs and 20 benchmarks illustrate the effectiveness of the proposed method.

**Strengths:**

- The proposed method tackles an important problem in VLMs to improve their inference efficiency.
- The proposed approach is shown effective on a variety of models and benchmarks.

**Weaknesses:**

- The introduction of IVC tokens brings the configurations of extra hyper-parameters such as setting token preservation ratios. It seems that the best settings depend on the underlying models, and it is unclear whether there can be a principled way to select these hyper-parameters.
- It would better if there can be more analyses and insights on the benefits of the proposed method; why they work better than others?
- The IVC seems to be derived by only considering Rm and Rn individually, but in ROPE, they are multiplied together for relative positional encoding. I'm not quite sure about the motivation here; there should be more illustration on this point (S3.2).

**Questions:**

(Please refer to the weakness part.)

---

> ### Author Response · Authors · 2025-11-22
> **Response to Question 1**
>
> We sincerely thank you for your time and constructive feedback. We are pleased that you recognize the importance of our work in improving VLM efficiency and find our proposed approach effective across various models and benchmarks. We will address your concerns below.
>
> > Q1: *The introduction of IVC tokens brings extra hyper-parameters such as token preservation ratios. The best settings appear to depend on the underlying model, and it is unclear whether there is a principled way to select these parameters.*
> >
>
> A1: Thank you for raising this important point. We agree that the token preservation ratio is a critical factor balancing efficiency and performance. Our observations align with yours that the optimal ratio depends on the model and the complexity of individual test samples. For instance, our empirical results indicate that while Qwen2.5-VL achieves comparable performance near a 50% ratio, LLaVA-v1.5 exhibits higher redundancy, maintaining robustness even at ~30%. Consequently, we fully recognize that a dynamic, task-adaptive  token pruning mechanism is a promising direction to address this variance.
>
> In the context of this paper, however, our preservation ratios followed the principle of fair comparison under a fixed budget. We standardized the pruning ratio (e.g., to ~50% in Tables 1 and 2) across all methods. This ensures that the observed performance differences are attributable solely to the effectiveness of the pruning strategies rather than the amount of information retained. As discussed in our Limitations (Appendix A.5 of the original paper) and Conclusion, we plan to extend IVC-Prune to integrate adaptive ratio selection in future work.

---

> ### Author Response · Authors · 2025-11-22
> **Response to Question 2**
>
> > Q2: More analyses and insights on the benefits of the proposed method and why it works better than others?
> >
>
> A2: Thank you for this opportunity to provide deeper insights into why IVC-Prune outperforms existing methods. We have provided a comprehensive analysis in Section 5.1 of the revised paper. Our advantages stem from three aspects:
>
> 1. Theoretical analysis of IVC tokens.
>
>     We present, to the best of our knowledge, the first theoretical analysis of the mathematical structure of RoPE in LVLMs, which reveals that certain token positions inherently serve as **Implicit Visual Coordinate (IVC) tokens.** These tokens are critical for enabling spatial reasoning. This perspective offers a novel insight into how LVLMs localize objects in images of arbitrary resolution. Experiments in Table 6 confirm that theoretically identified IVC tokens lead to markedly better performance than heuristic‑based selection rules, validating the value of our analysis.
>
> 2. Robust and architecture‑agnostic token selection.
>
>     Our two‑stage foreground token selection method leverages value-vector similarity, followed by semantic seeding and contextual refinement. This design effectively reduces positional bias from attention distributions and demonstrates strong generalization ability across architectures.  We observe consistent gains on four distinct LVLM families over twenty public benchmarks, whereas many prior works focus narrowly on a single architecture (often LLaVA), limiting their transferability.
>
> 3. Efficiency of IVC-Prune strategy.
>
>     We design a token pruning strategy that safely applies pruning to **early‑layer KV caches**, achieving **maximal KV‑cache reduction** and improved decoding efficiency. To our knowledge, previous methods avoided early‑layer pruning because it often caused severe degradation. Our theoretical understanding of IVC tokens allows us to overcome this limitation, thus achieving both high pruning rates and stable accuracy.
>
>
> Overall, IVC‑Prune is built on a formal analysis of LVLM spatial reasoning, which in turn informs a robust, architecture‑agnostic token selection strategy and a safe early‑layer pruning mechanism.

---

> ### Author Response · Authors · 2025-11-22
> **Response to Question 3**
>
> > Q3: The IVC seems to be derived by only considering Rm and Rn individually, but in ROPE, they are multiplied together for relative positional encoding. What is the motivation for this approach?
> >
>
> A3: Thank you for this insightful question. We agree with your observation that RoPE encodes relative positions through the product $R_{-n}R_{m}= R_{m-n}$ (Eq. (3)). Our analysis does not contradict this fundamental property; rather, we investigate how absolute spatial awareness in the image tokens emerges from this relative encoding mechanism.
>
> The core motivation lies in the challenge of spatial reasoning in LVLMs. Since visual tokens are flattened into 1D sequences, often with dynamic resolutions, the model lacks an explicit 2D grid structure. To perform tasks like visual grounding, the model must recover absolute positions from relative positional encoding.
>
> Our insight is that absolute positions can be derived from relative distances if specific tokens serve as fixed references. Mathematically, the attention score involves the term $ R_{-n} R_{m} $. We analyze the specific conditions of key positions $m$ that allow the query $ n$ to resolve its own absolute location:
>
> 1. When $ R_m$ is close to the identity matrix $I$, the relative term becomes $ R_{m-n} ≈ R_{-n}I =  R_{-n}$. By attending to these tokens, the query effectively "sees" its own absolute rotational state relative to a real-axis reference.
> 2. When $R_m$ is close to the $90^\circ$ rotation matrix $ J$, the term becomes  $R_{m-n} = R_{-n}J$. This provides an imaginary-axis reference.
>
> Therefore, analyzing $R_{m}$ individually allows us to identify which tokens act as these special tokens. These tokens effectively work as an implicit coordinate system, enabling other tokens to orient themselves via the standard attention mechanism.

---

### Official Review · Reviewer_4Zjm · 2025-10-30

**Soundness:** 3
**Presentation:** 3
**Contribution:** 2
**Rating:** 4
**Confidence:** 4

**Summary:**

This paper introduces IVC-Prune, a training-free, text-aware pruning approach for LVLM inference accelaration. Specifically, IVC-Prune identifies some reference tokens from implicit visual coordinates, and presearves those IVC tokens and foreground tokens for effective reasoning. Experimental results on standard benchmakes across four widely used LVLMs validate the effectiveness of the proposed approach.

**Strengths:**

1. The proposed approach seems to be simple yet effective and can be applied to LVLMs with different architectures.
2. Extensive experimental results validate the effectiveness of the proposed approach.
3. The paper is generally well-written and the stucture is clear.

**Weaknesses:**

1. While the empirical experiments demonstrate the significant impact of the IVC tokens, I would be more convinced if a more detailed analysis were provided to explain *why* these tokens are important.
2. Relevant baselines employing window-based token selection approaches should be included in the main experiments, as they also aim to preserve spatial information along with the foreground tokens. Additionally, a discussion on novelty is needed to better differentiate this work from those related methods.
3. The foreground token selection strategy seems to primarily combine existing methods rather than introducing fundamentally new ideas.
4. I would like to see an evaluation of the proposed approach under extremely low token budgets (e.g., 11.1%–33.3%), as reported in prior work [3], to better understand its robustness.

[1] Token Pruning in Multimodal Large Language Models: Are We Solving the Right Problem? https://www.arxiv.org/abs/2502.11501

[2] VScan: Rethinking Visual Token Reduction for Efficient Large Vision-Language Models. https://arxiv.org/abs/2505.22654

[3] PyramidDrop: Accelerating Your Large Vision-Language Models via Pyramid Visual Redundancy Reduction. https://arxiv.org/abs/2410.17247

**Questions:**

See the weaknesses mentioned above. I am open to updating my rating if they are properly addressed.

---

> ### Author Response · Authors · 2025-11-22
> **Response to Question 1**
>
> We sincerely thank you for your constructive review. We are encouraged that you find our proposed IVC-Prune to be 'simple yet effective' and applicable across 'different architectures.' We also appreciate your recognition of our extensive experiments and clear presentation. Below, we will address the concerns you've raised and offer further clarification.
>
> > Q1: Why are IVC tokens specifically important?
> >
>
> A1: Thank you for raising this important point. We appreciate the opportunity to clarify the theoretical analysis of *why* IVC tokens are crucial for spatial reasoning, as this is a core contribution of our paper. Below we summarize the theoretical analysis (Sections 3.1 and 3.2) and the empirical evidence that supports it.
>
> Our analysis starts from a fundamental property of RoPE (Eq. (3)): it inherently encodes **relative positions ($m-n$)** via structured rotations, $R_{m-n} = R_{-n} R_{m}$, in the attention mechanism. However, spatial reasoning tasks (e.g. grounding)  require the model to understand the absolute positions in arbitrary resolution images. This raises the central question: *How does the model know ”where” an object is in an arbitrary resolution image with relative positions?* Our insight is that LVLMs implicitly learn to use specific tokens as **an** **implicit coordinate system (IVC tokens).** By attending to these tokens, the model can recover absolute coordinates from relative encodings.
>
> This leads to the next question: Which positions serve as effective coordinate tokens? We hypothesize that ideal candidates are positions $ m $ where the RoPE rotation matrix $R_{m}$  approximates a canonical transformation. Specifically, the relative position term in attention is  $ R_{m-n} = R_{-n} R_{m} $ . If $R_{m}$ approximates the identity matrix ($R_{m} ≈ I$), this term simplifies to $ R_{-n} $, effectively isolating the query’s absolute positional component. Based on this, we identified two such transformations:
>
> 1. **The Identity Matrix (I):** This corresponds to a real-axis reference. As shown in Eq. (4) and Eq. (5), minimizing the distance $\|R_{m} - I\|_F^2$ is equivalent to maximizing the sum of cosine components $V(m)$.
> 2. **The 90° Rotation Matrix (J):** This provides an imaginary-axis reference. As shown in Eq. (6) and Eq. (7), minimizing $\|R_m - J\|_F^2$ is equivalent to maximizing the sum of sine components $U(m)$.
>
> Therefore, IVC tokens are the mathematically determined positions that allow the model to establish implicit coordinate systems.
>
> This theoretical insight is validated by our ablation studies. We demonstrated that adding back IVC tokens in FastV and PDrop allows the model to recover its spatial reasoning capabilities (Table 4).  In Table 11 and Figure 1, we demonstrate that restricting inputs to foreground tokens leads to a substantial performance drop. Remarkably, augmenting these with just 10% IVC tokens often allows the model to surpass the unpruned baseline. Furthermore, to prove that this benefit arises from the specific RoPE properties rather than general geometric distribution, we compared IVC tokens against heuristic patterns like corners or diagonals (Table 6). The significant superiority of IVC tokens confirms our hypothesis: LVLMs rely specifically on these mathematically unique positions to ground their spatial understanding.

---

> > ### Author Response · Authors · 2025-11-22
> > **Response to Question 3 and 4**
> >
> > > Q3:  Additional discussion on novelty between this work and related methods.
> > >
> >
> > A3: Thank you for this suggestion. We have revised Section 5.1 of the revised manuscript to better clarify our novelty. Our approach advances prior LVLM pruning methods in the following aspects:
> >
> > 1. **First theoretical analysis of spatial reasoning in LVLMs.**
> >
> >     We provide the first theoretical analysis of the *mathematical structure of RoPE* in LVLMs, revealing that certain token positions inherently serve as **Implicit Visual Coordinate (IVC) tokens** crucial for absolute spatial reasoning. This provides novel mechanistic insight into how LVLMs localize objects at arbitrary resolution, an aspect not explored in earlier pruning work.
> >
> > 2. **New explanation for early-layer pruning sensitivity.**
> >
> >     Prior studies [3, 4] observed that pruning early layers causes significant performance drops, without clarifying the cause. Our experiments (Table 6, Table 11) reveal that the main factor is the removal of IVC tokens rather than pruning itself. This not only explains the root cause but also offers a clear strategy to avoid the issue.
> >
> > 3. **Enabling efficient pruning of early-layer KV caches.**
> >
> >     Guided by our IVC token analysis, we develop a pruning strategy that safely prunes **early-layer KV caches** while preserving performance. While many existing training-free methods decide to preserve early layer KV caches to ensure stability, our method demonstrates that early-layer KV caches can be pruned efficiently if the spatial tokens are preserved, leading to greater memory savings.
> >
> > 4. **Enhanced cross-architecture robustness.**
> >
> >     We introduce a **two-stage semantic foreground selection method** designed to mitigate positional bias in attention scores. While some prior methods are optimized for specific architectures (often LLaVA), our strategy shows consistent effectiveness across **four diverse LVLM** **families** and **twenty benchmarks**. This suggests that our approach is robust and adaptable to various LVLMs.
> >
> >
> > In summary, we believe that our work contributes a valuable theoretical foundation for spatial reasoning in LVLMs, alongside an architecture-agnostic pruning strategy that is both efficient and robust. These distinctive aspects differentiate IVC-Prune from prior studies, advancing the field in terms of both theoretical analysis and practical application.
> >
> > [3] PyramidDrop: Accelerating Your Large Vision-Language Models via Pyramid Visual Redundancy Reduction. https://arxiv.org/abs/2410.17247v2
> >
> > [4] ATP-LLaVA: Adaptive Token Pruning for Large Vision Language Models. https://arxiv.org/abs/2412.00447
> >
> >
> > > Q4: The foreground token selection strategy seems to primarily combine existing methods rather than introducing fundamentally new ideas.
> > >
> >
> > A4: Thank you for this comment. We would like to highlight that the fundamental contribution of our work is the theoretical analysis of IVC tokens. The foreground token selection strategy serves as a robust practical implementation designed to maximize the efficacy of this theoretical framework.
> >
> > While our strategy incorporates established concepts, its specific design and robust application offer novel advantages. Although value‐vector similarity has been explored in prior work [5], our approach differs both in motivation and application. Specifically, we do not employ it merely as an alternative similarity metric; rather, we strategically leverage value vectors to mitigate the positional bias inherent in raw attention scores. Moreover, our method introduces a two-stage design (semantic seeding followed by foreground refinement). This design allows the method to adapt to variable image inputs and attention patterns in different LVLMs.  Our strategy demonstrates robustness across different LVLM families, while many prior methods are tailored to specific models.
> >
> > [5] FlowCut: Rethinking Redundancy via Information Flow for Efficient Vision-Language Models. https://arxiv.org/abs/2505.19536

---

> ### Author Response · Authors · 2025-11-22
> **Response to Question 2 (part1/2)**
>
> > Q2: Relevant baselines employing window-based token selection approaches should be included in the main experiments.
> >
>
> A2: We appreciate this valuable suggestion. We agree that including window-based token selection methods is essential for a comprehensive evaluation.  Following your recommendation, we have conducted additional experiments with Window FastV [1] and VScan [2].
>
> We implemented Window FastV for dynamic-resolution inputs, adapting the algorithm described in [1]. This method selects foreground tokens via attention and supplements them with tokens from a fixed window to preserve spatial structure.
>
> We successfully reproduced VScan [2] on Qwen2.5-VL and LLaVA-v1.5. Since the original VScan codebase does not support InternVL 2.5 and DeepSeek‑VL2, and adapting it to complex dynamic‑resolution tiling for these models requires substantial engineering,  we are still working on extending VScan to InternVL2.5 and DeepSeek-VL2.
>
> Table R-10: Additional baselines on visual grounding benchmarks. * indicates added results.
>
> | Models | Method | Average Tokens | RefCOCO testA | RefCOCO testB | RefCOCO val | RefCOCO+ testA | RefCOCO+ testB | RefCOCO+ val | RefCOCOg test | RefCOCOg val | Relative Avergae |
> | --- | --- | --- | --- | --- | --- | --- | --- | --- | --- | --- | --- |
> | Qwen2.5-VL 7B | Vanilla | 100% | 92.2  | 84.7  | 89.6  |  88.0  | 74.3  | 82.8  | 86.9  | 86.8 | 100% |
> |  | FastV | 54% | 74.4  | 76.5  | 75.4  | 68.9  | 66.8  | 67.7  | 75.3  | 74.8 | 84.7% |
> |  | *Window FastV | 54% | 82.9 | 79.8 | 81.8 | 77.4 | 69.0 | 74.0 | 79.2 | 79.5 | 91.0% |
> |  | *VScan | 50% | 90.2 | 82.2 | 86.7 | 84.6 | 70.6 | 79.0 | 83.6 | 83.9 | 96.4% |
> |  | IVC-Prune | 50% | 92.0  | 84.5  | 89.3  | 87.4  | 74.1  | 82.4  | 86.5  | 86.5 | 99.6% |
> | InternVL 2.5 8B | Vanilla | 100%  | 94.7  | 86.0  | 90.3 |  91.5  | 78.7  | 85.1  | 87.6 |  87.1  | 100% |
> |  | FastV | 53% |  87.0 |  77.6  | 81.6  | 82.6  | 70.7  | 76.1  | 77.9  | 78.5  | 90.1% |
> |  | *Window FastV | 53% | 82.9 | 73.6 | 78.8 | 80.1 | 66.6 | 73.5 | 74.4 | 73.4 | 86.0% |
> |  | IVC-Prune | 50%  | 94.2  | 85.7  | 90.3  | 91.1  | 78.2  | 84.8  | 86.9  | 86.4  | 99.5% |
> | DeepSeek-VL2  Small | Vanilla | 100%  | 96.5  | 92.6  | 95.2  | 94.7  | 87.9  | 91.4 |  93.3  | 93.2  | 100% |
> |  | FastV | 54% |  94.4  | 89.5 |  92.6  | 91.8 |  83.6  | 87.8  | 90.6  | 90.4  | 96.7% |
> |  | *Window FastV | 54% | 95.0 | 90.4 | 93.6 | 92.5 | 85.2 | 89.2 | 91.3 | 90.9 | 97.8% |
> |  | IVC-Prune | 52%  | 96.0  | 91.8  | 94.5  | 94.0  | 86.6  | 90.3  | 92.4  | 92.2  | 99.0% |
>
> Table R-11: Addition baselines on general VQA benchmarks .
>
> | Models | Method | Average Tokens | SEED | MMB | MMS | RWQA | MME | POPE | HallB | TVQA | AI2D | Relative Avergae |
> | --- | --- | --- | --- | --- | --- | --- | --- | --- | --- | --- | --- | --- |
> | Qwen2.5VL 7B | Vanilla | 100% | 76.7  | 82.4 |  64.2  | 67.8  | 2310.6  | 86.9  | 51.5  | 84.9  | 83.8 | 100% |
> |  | FastV | 54% | 72.9  | 80.5  | 59.8  | 68.5  | 2242.5 |  86.2  | 54.3  | 84.7  | 81.6  | 98.4% |
> |  | *Window FastV | 54% | 73.9 | 80.6 | 58.1 | 67.4 | 2235.5 | 85.9 | 49.4 | 83.9 | 81.8 | 96.9% |
> |  | *VScan | 50% | 74.8 | 80.6 | 59.9 | 68.4 | 2285.0 | 87.3 | 56.5 | 84.3 | 79.1 | 99.1% |
> |  | IVC-Prune | 50% | 76.7 |  82.6  | 62.9  | 68.2 |  2303.1  | 87.6  | 54.8 |  84.4  | 84.2 | 99.6% |
> | InternVL 2.5 8B | Vanilla | 100% | 77.1  | 83.2 |  62.7 |  69.4  | 2344.0  | 89.0  | 50.8  | 79.0  | 84.4 | 100% |
> |  | FastV | 53% | 74.0  | 81.6  | 62.5  | 65.0 |  2268.3  | 86.7  | 48.8  | 76.8  | 83.1  | 97.0% |
> |  | *Window FastV | 53% | 73.9 | 82.0 | 57.7 | 65.8 | 2254.0 | 87.1 | 48.3 | 76.3 | 82.8 | 96.1% |
> |  | IVC-Prune | 50% | 77.0 |  83.0 | 62.6 |  69.9  | 2308.2  | 88.9  | 50.2  | 78.0  | 84.3  | 99.6% |
> | DeepSeek-VL2Small-16B | Vanilla | 100% | 76.9 |  79.2  | 57.7  | 70.3  | 2128.6 |  89.3  | 43.8  | 83.4  | 82.0 | 100% |
> |  | FastV | 54% | 75.6 |  78.2 |  55.9 |  69.0 |  2112.8  | 89.2  | 42.7 |  83.1  | 81.0  | 98.6% |
> |  | *Window FastV | 54% | 76.1 | 78.3 | 56.2 | 68.2 | 2122.4 | 89.0 | 38.1 | 82.3 | 80.5 | 97.3% |
> |  | IVC-Prune | 52% | 77.0  | 79.3 |  57.7  | 70.3  | 2132.2  | 89.5  | 44.3  | 83.0  | 81.8 | 100.1% |
> | LLaVA-v1.5 7B | Vanilla | 100% | 64.4  | 60.6  | 34.2  | 54.5  | 1543.1  | 74.5  | 25.8  | 20.7  | 49.1  | 100% |
> |  | FastV | 30% | 60.1  | 59.8  | 33.5  | 50.2  | 1555.2  | 73.4  | 28.6  | 21.0  | 49.0  | 99.3% |
> |  | *Window FastV | 30% | 62.2 | 60.2 | 34.1 | 51.6 | 1643.3 | 78.2 | 27.4 | 19.8 | 48.8 | 100.3% |
> |  | *VScan | 30% | 63.9 | 60.5 | 32.6 | 51.8 | 1637.5 | 78.8 | 28.0 | 21.0 | 49.0 | 101.2% |
> |  | IVC-Prune | 28% | 64.4  | 60.6  | 34.5  | 54.5  | 1554.4  | 77.6  | 26.7  | 21.1  | 49.2  | 101.3% |

---

> > ### Author Response · Authors · 2025-11-22
> > **Response to Question 2 (part 2/2)**
> >
> > We further evaluated the **window sampling pattern** in our pattern comparison in Table R-12. While window sampling preserves local geometry, it is still a heuristic. In contrast, IVC tokens serve as global *coordinate anchors* derived from the mathematical properties of RoPE. This confirms that mathematically precise IVC tokens are more effective than generic local windows. We have updated Tables R-10 and R-11 in the main experiments and Table R-12 in Table 6.
> >
> > Table R-12: Ablation study comparing IVC tokens with alternative visual token patterns.
> >
> > |  | None | Random | C Points | ***Window** | Diagonal | IVC 5% | IVC 10% | IVC 20% | Baseline |
> > | --- | --- | --- | --- | --- | --- | --- | --- | --- | --- |
> > | RefCOCO testA | 58.0 | 79.3 | 73.5 | **89.0** | 89.8 | 89.1 | 92.8 | 93.3 | 92.2 |
> > | RefCOCO+ testA | 56.8 | 77.6 | 71.8 | **86.3** | 87.0 | 87.0 | 90.0 | 90.4 | 88.0 |
> > | GQA choose all | 90.3 | 92.0 | 91.3 | **93.3** | 92.3 | 93.3 | 93.3 | 93.7 | 93.7 |
> >
> > [1] Token Pruning in Multimodal Large Language Models: Are We Solving the Right Problem? https://www.arxiv.org/abs/2502.11501
> >
> > [2] VScan: Rethinking Visual Token Reduction for Efficient Large Vision-Language Models. https://arxiv.org/abs/2505.22654

---

> ### Author Response · Authors · 2025-11-22
> **Response to Question 5**
>
> > Q5: Additional results at extremely low token budgets (e.g., 11.1%–33.3%), as reported in PDrop.
> >
>
> A5: Thank you for the suggestion. Following the evaluation protocol in PDrop, we conducted a new set of experiments on LLaVA-v1.5 7B under the extremely low token budgets suggested (33.3%, 22.2%, and 11.1%). We have included these detailed results in Appendix Section A.7 of the revised paper.
>
> IVC-Prune demonstrates remarkable robustness even under aggressive token reduction. At 11.1% token retention, our method achieves 97.2% relative performance.
>
> Implementation Note: Regarding the comparative baseline PDrop, as the official configuration for these specific low-budget settings was not explicitly documented in the original paper, we adopted the configurations recommended in the official repository's discussion threads (Issues #29, #10). Specifically, we set the pruning layers to {2, 8, 16} and adjusted *λ* to 0.56, 0.42, and 0.19 to align with the target budgets.
>
> [Issues #29] https://github.com/Cooperx521/PyramidDrop/issues/29
>
> [Issues #10] https://github.com/Cooperx521/PyramidDrop/issues/10
>
> Table R-13: Performance comparison under extremely low token budgets on LLaVA-v1.5 7B
>
> | Method | Aveage Tokens | SEEDBench | MMBench | MMstar | RealWorldQA | MME | POPE | HallusionBench | TextVQA_VAL | AI2D_TEST | Rel. Avg. |
> | --- | --- | --- | --- | --- | --- | --- | --- | --- | --- | --- | --- |
> | Vanilla | 576 (100%) | 64.4 | 60.6 | 34.2 | 54.5 | 1543.1 | 74.5 | 25.8 | 20.7 | 49.1 | 100% |
> | FastV | 173 (30%) | 60.1 | 59.8 | 33.5 | 50.2 | 1555.2 | 73.4 | 28.6 | 21.0 | 49.0 | 99.3% |
> | PDrop | 271 (47%) | 63.6 | 60.2 | 33.6 | 53.7 | 1600.4 | 79.5 | 28.0 | 18.9 | 49.4 | 100.6% |
> | IVC-Prune | 161 (28%) | 64.4 | 60.6 | 34.5 | 54.5 | 1554.4 | 77.6 | 26.7 | 21.1 | 49.2 | 101.3% |
> |  |  |  |  |  |  |  |  |  |  |  |  |
> | FastV | 192 (33.3%) | 61.2 | 60.2 | 33.4 | 51.6 | 1572.7 | 74.8 | 29.0 | 21.8 | 48.8 | 100.7% |
> | PDrop | 192 (33.3%) | 60.0 | 54.6 | 31.9 | 51.2 | 1607.6 | 80.1 | 25.7 | 17.0 | 48.6 | 95.9% |
> | IVC-Prune | 192 (33.3%) | 64.5 | 60.7 | 34.5 | 54.4 | 1567.7 | 76.9 | 26.2 | 21.1 | 49.1 | 101.0% |
> | FastV | 128 (22.2%) | 57.2 | 58.3 | 33.1 | 47.7 | 1462.1 | 67.8 | 28.1 | 18.7 | 48.0 | 94.7% |
> | PDrop | 128 (22.2%) | 56.0 | 49.8 | 31.1 | 50.8 | 1565.4 | 77.2 | 24.4 | 14.8 | 45.1 | 90.7% |
> | IVC-Prune | 128 (22.2%) | 64.5 | 60.7 | 34.0 | 54.4 | 1525.6 | 77.0 | 26.2 | 20.1 | 49.1 | 100.0% |
> | FastV | 64 (11.1%) | 45.8 | 40.2 | 28.4 | 38.2 | 1109.7 | 31.9 | 27.1 | 6.6 | 45.8 | 70.6% |
> | PDrop | 64 (11.1%) | 46.3 | 39.2 | 28.3 | 46.5 | 1205.7 | 51.0 | 23.6 | 8.6 | 46.2 | 75.4% |
> | IVC-Prune | 64 (11.1%) | 64.2 | 60.4 | 33.9 | 54.1 | 1511.3 | 75.4 | 24.8 | 16.9 | 49.1 | 97.2% |

---

> > ### Comment · Reviewer_4Zjm · 2025-11-22
> >
> > Thanks for your detailed responses, which have addressed all of my concerns. I'm updating the rating to 6 to recommend acceptance.

---

> > > ### Author Response · Authors · 2025-11-22
> > >
> > > We are glad to hear that our responses have addressed your concerns and questions. Thank you for carefully reviewing our comments and providing constructive feedback. We have incorporated all suggested revisions and clarifications into the updated version of the manuscript. Thank you once again for your valuable, constructive feedback and for your consideration.

---

### Official Review · Reviewer_pnux · 2025-11-01

**Soundness:** 3
**Presentation:** 3
**Contribution:** 3
**Rating:** 8
**Confidence:** 4

**Summary:**

This paper proposed a new method for visual token pruning in VLMs. Starting with a theoretical analysis of ROPE embeddings, the authors pointed out that ROPE implicitly encode spatial information already when processing visual tokens. In particular, at positions where the rotation matrix is close to an identity matrix or 90 degree rotation matrix, the model can learn to use those positions as anchors to better understand the visual inputs. Based on this observation, they proposed always keep visual tokens at these positions during pruning, and further keep semantically salient tokens to maximize task performance (that are computed based on similarity between value vectors instead of attention scores, to minimize the impact of positional bias).  The actual pruning is determined at an intermediate layer in VLM, and once the tokens are decides, the entire layers’ KV cache for those pruned tokens are removed. The experiments with 4 different VLMs on a wide range of visual understanding tasks showcase the effectiveness of the proposed approach.

**Strengths:**

Unlike most previous methods on this area, this paper actually conducted a nice theoretically analysis of the working mechanism in VLMs, and the strong empirical results further supported the analysis.

The pruning strategy is well designed, once the tokens for pruning are decided in the first forward pass, all layers KV cache can be cleaned to maximize the saving in compute and memory.

The experiments with 4 different VLMs with different architectures, image handling strategies all show very strong performance across a wide range of tasks (including visual grounding, fine-grained perception tasks, which are often overlooked), clearly showing the advantage of the proposed method.

The ablations are well done to verify the effectiveness of IVC tokens, and explains well why previous works might have different findings → the IVC tokens are pruned in the earlier layers.

**Weaknesses:**

I don’t have any major concerns. One minor issue is that the methods depends on the property of RoPE, thus the generalizability to other model architectures with different position embeddings is unknown.

**Questions:**

None

---

> ### Author Response · Authors · 2025-11-22
>
> Thank you for your thoughtful and positive feedback! We are sincerely encouraged by your recognition of our work. We deeply appreciate your comments highlighting the soundness of our theoretical analysis on RoPE and the effectiveness of our pruning strategy.
>
> Regarding your minor concern about generalizability to non-RoPE architectures: We acknowledge that our theoretical foundation is specifically tailored to RoPE (as noted in Section A.5, Limitation, of the original manuscript). However, we would like to highlight that RoPE and its variants have become the de facto standard for state-of-the-art LVLMs.  Our experiments demonstrate that IVC-Prune is robust not only to standard 2D-RoPE (employed in DeepSeek-VL2 and LLaVA v1.5), but also to more complex variants such as Multimodal RoPE (used in Qwen2.5-VL) and Dynamic NTK-Scaling RoPE (used in InternVL 2.5). This empirical success across diverse RoPE implementations suggests that our method is widely applicable to current LVLMs.
>
> Thank you once again for your valuable time and support.

---

### Official Review · Reviewer_9fb3 · 2025-11-01

**Soundness:** 3
**Presentation:** 3
**Contribution:** 3
**Rating:** 6
**Confidence:** 4

**Summary:**

The paper proposes a novel token pruning strategy, called IVC-Prune. The central idea of the paper is that LVLMs implicitly establish visual coordinate systems through RoPE, and certain token positions act as implicit visual coordinates (IVC tokens) that are crucial for spatial reasoning tasks. The authors propose a training-free, prompt-aware pruning strategy that retains both IVC tokens and semantically relevant foreground tokens. This method significantly reduces the number of visual tokens while preserving or even improving performance on various vision tasks.

**Strengths:**

1. The paper introduces a new perspective on token pruning by focusing on the IVC tokens for spatial reasoning in LVLMs. The idea of implicit visual coordinate from RoPE is novel.
2. The experiment results are impressive. The performance can approach or even surpass the vanilla model under 50% compression.

**Weaknesses:**

1. The paper lacks comparison with more recent baselines. FastV is already an weaker baseline, and it would be beneficial to include a comparison with SparseVLM.
2. The paper does not clarify how IVC tokens and foreground tokens should be allocated under a 50% total budget. Ablation experiments should be included to investigate this.

**Questions:**

1 & 2: Please see the weakness.
3. Tables 3 and 5 do not include data for PDrop.
4. It is recommended to add results on high-resolution benchmarks, such as DocVQA and OCRBench, as these benchmarks involve a larger number of visual tokens.

---

> ### Author Response · Authors · 2025-11-22
> **Response to Question 1 (part 1/2)**
>
> Thank you for your insightful comments and recognition of our work. We greatly appreciate your recognition of the novelty of our IVC tokens and the impressive experimental results achieved by IVC-Prune. Below, we will address your concerns and provide further clarification.
> > Q1: Lack of comparison with more recent baselines such as SparseVLM.
> >
> A1: Thank you for this valuable suggestion. We have conducted additional experiments to provide a comprehensive comparison with recent baselines, including SparseVLM and other state-of-the-art methods.
>
> 1. **Analysis of SparseVLM.** We reproduced SparseVLM on Qwen2.5-VL 7B with two variants, as shown in Table R-1 and R-2. Our experiments reveal a **fundamental limitation** of SparseVLM for spatial reasoning tasks: **Position ID Reconstruction Issue.** SparseVLM's token clustering mechanism requires reconstructing position IDs for merged tokens, which inherently discards original spatial location information. This is critical for tasks requiring precise spatial reasoning, such as visual grounding. The default SparseVLM (with clustering) suffered severe performance degradation on grounding benchmarks (dropping to 15.3%), while the variant without clustering achieved 76.1%. On general VQA benchmarks, however, SparseVLM with clustering performs comparably to baseline (99.6%).
>
>     This aligns with the observations in [1] (Section 2, lines 136-138 of the original manuscript). We have added a dedicated analysis in Appendix Section A.5 of the revised paper to provide a more comprehensive discussion.
>
>
> Table R-1. Variants of SparseVLM on visual grounding benchmarks.
>
> | Qwen2.5-VL 7B | Average Tokens | RefCOCO testA | RefCOCO testB | RefCOCO val | RefCOCO+ testA | RefCOCO+ testB | RefCOCO+ val | RefCOCOg test | RefCOCOg val | Relative Avergae |
> | --- | --- | --- | --- | --- | --- | --- | --- | --- | --- | --- |
> | Vanilla | 100% | 92.2 | 84.7 | 89.6 | 88.0 | 74.3 | 82.8 | 86.9 | 86.8 | 100% |
> | IVC-Prune | 50% | 92.0 | 84.5 | 89.3 | 87.4 | 74.1 | 82.4 | 86.5 | 86.5 | 99.6% |
> | SparseVLM w/o clustering | 49% | 77.2 | 61.2 | 69.1 | 71.7 | 53.5 | 63.2 | 63.2 | 63.6 | 76.1% |
> | SparseVLM w clustering (default) | 51% | 14.1 | 13.8 | 14.2 | 12.6 | 11.7 | 12.5 | 12.3 | 13.6 | 15.3% |
>
> Table R-2. Variants of SparseVLM on general VQA benchmarks.
> | Qwen2.5-VL 7B | Average Tokens | SEED | MMB | MMS | RWQA | MME | POPE | HallB | TVQA | AI2D | Relative Avergae |
> | --- | --- | --- | --- | --- | --- | --- | --- | --- | --- | --- | --- |
> | Vanilla | 100% | 76.7 | 82.4 | 64.2 | 67.8 | 2310.6 | 86.9 | 51.5 | 84.9 | 83.8 | 100% |
> | IVC-Prune | 50% | 76.7 | 82.6 | 62.9 | 68.2 | 2303.1 | 87.6 | 54.8 | 84.4 | 84.2 | 100.6% |
> | SparseVLM w/o clustering | 49% | 74.9 | 79.6 | 45.6 | 61.4 | 2279.6 | 85.9 | 53.5 | 83.7 | 82.3 | 94.9% |
> | SparseVLM w clustering (default) | 51% | 74.8 | 82.0 | 61.3 | 67.8 | 2320.0 | 86.8 | 54.5 | 84.5 | 82.5 | 99.6% |

---

> > ### Author Response · Authors · 2025-11-22
> > **Response to Question 1 (part 2/2)**
> >
> > 2. **Comparison with Additional Recent Baselines.** To further strengthen baseline coverage, we added comparisons with **Window FastV** [2] and **VScan** [3]. Since many recent methods are implemented specifically for LLaVA, adapting them to modern architectures (e.g., Qwen2.5-VL, InternVL2.5, DeepSee-VL2) requires substantial engineering effort. For VScan, official code was not available for InternVL2.5 and DeepSeek-VL2 at the time, and reproduction work is still in progress. The results in Tables R-3 and R-4 demonstrate that IVC-Prune consistently outperforms these recent baselines. We have included these comparisons in the revised manuscript.
> > Table R-3. Additional baselines on visual grounding benchmarks. * indicates added results.
> >
> > |  |  | Average Tokens | RefCOCO testA | RefCOCO testB | RefCOCO val | RefCOCO+ testA | RefCOCO+ testB | RefCOCO+ val | RefCOCOg test | RefCOCOg val | Relative Avergae |
> > | --- | --- | --- | --- | --- | --- | --- | --- | --- | --- | --- | --- |
> > | Qwen2.5-VL 7B | Vanilla | 100% | 92.2 | 84.7 | 89.6 | 88.0 | 74.3 | 82.8 | 86.9 | 86.8 | 100% |
> > |  | IVC-Prune | 50% | 92.0 | 84.5 | 89.3 | 87.4 | 74.1 | 82.4 | 86.5 | 86.5 | 99.6% |
> > |  | *Window FastV | 54% | 82.9 | 79.8 | 81.8 | 77.4 | 69.0 | 74.0 | 79.2 | 79.5 | 91.0% |
> > |  | *VScan | 50% | 90.2 | 82.2 | 86.7 | 84.6 | 70.6 | 79.0 | 83.6 | 83.9 | 96.4% |
> > | InternVL 2.5 8B | Vanilla | 100%  | 94.7  | 86.0  | 90.3 |  91.5  | 78.7  | 85.1  | 87.6 |  87.1  | 100% |
> > |  | IVC-Prune | 50%  | 94.2  | 85.7  | 90.3  | 91.1  | 78.2  | 84.8  | 86.9  | 86.4  | 99.5% |
> > |  | *Window FastV | 53% | 82.9 | 73.6 | 78.8 | 80.1 | 66.6 | 73.5 | 74.4 | 73.4 | 86.0% |
> > | DeepSeek-VL2  Small | Vanilla | 100%  | 96.5  | 92.6  | 95.2  | 94.7  | 87.9  | 91.4 |  93.3  | 93.2  | 100% |
> > |  | IVC-Prune | 52%  | 96.0  | 91.8  | 94.5  | 94.0  | 86.6  | 90.3  | 92.4  | 92.2  | 99.0% |
> > |  | *Window FastV | 54% | 95.0 | 90.4 | 93.6 | 92.5 | 85.2 | 89.2 | 91.3 | 90.9 | 97.8% |
> >
> > Table R-4. Additional baselines on general VQA benchmarks.
> >
> > |  |  | Average Tokens | SEED | MMB | MMS | RWQA | MME | POPE | HallB | TVQA | AI2D | Relative Avergae |
> > | --- | --- | --- | --- | --- | --- | --- | --- | --- | --- | --- | --- | --- |
> > | Qwen2.5-VL 7B | Vanilla | 100% | 76.7 | 82.4 | 64.2 | 67.8 | 2310.6 | 86.9 | 51.5 | 84.9 | 83.8 | 100% |
> > |  | IVC-Prune | 50% | 76.7 | 82.6 | 62.9 | 68.2 | 2303.1 | 87.6 | 54.8 | 84.4 | 84.2 | 100.6% |
> > |  | *Window FastV | 54% | 73.9 | 80.6 | 58.1 | 67.4 | 2235.5 | 85.9 | 49.4 | 83.9 | 81.8 | 96.9% |
> > |  | *VScan | 50% | 74.8 | 80.6 | 59.9 | 68.4 | 2285.0 | 87.3 | 56.5 | 84.3 | 79.1 | 99.1% |
> > | InternVL 2.5 8B | Vanilla | 100% | 77.1  | 83.2 |  62.7 |  69.4  | 2344.0  | 89.0  | 50.8  | 79.0  | 84.4 | 100% |
> > |  | IVC-Prune | 50% | 77.0 |  83.0 | 62.6 |  69.9  | 2308.2  | 88.9  | 50.2  | 78.0  | 84.3  | 99.6% |
> > |  | *Window FastV | 53% | 73.9 | 82.0 | 57.7 | 65.8 | 2254.0 | 87.1 | 48.3 | 76.3 | 82.8 | 96.1% |
> > | DeepSeek-VL2  Small | Vanilla | 100% | 76.9 |  79.2  | 57.7  | 70.3  | 2128.6 |  89.3  | 43.8  | 83.4  | 82.0 | 100% |
> > |  | IVC-Prune | 52% | 77.0  | 79.3 |  57.7  | 70.3  | 2132.2  | 89.5  | 44.3  | 83.0  | 81.8 | 100.1% |
> > |  | *Window FastV | 54% | 76.1 | 78.3 | 56.2 | 68.2 | 2122.4 | 89.0 | 38.1 | 82.3 | 80.5 | 97.3% |
> > | LLaVA-v1.5 7B | Vanilla | 100% | 64.4  | 60.6  | 34.2  | 54.5  | 1543.1  | 74.5  | 25.8  | 20.7  | 49.1  | 100% |
> > |  | IVC-Prune | 28% | 64.4  | 60.6  | 34.5  | 54.5  | 1554.4  | 77.6  | 26.7  | 21.1  | 49.2  | 101.3% |
> > |  | *Window FastV | 30% | 62.2 | 60.2 | 34.1 | 51.6 | 1643.3 | 78.2 | 27.4 | 19.8 | 48.8 | 100.3% |
> > |  | *VScan | 30% | 63.9 | 60.5 | 32.6 | 51.8 | 1637.5 | 78.8 | 28.0 | 21.0 | 49.0 | 101.2% |
> >
> > [1] Grounding-Aware Token Pruning: Recovering from Drastic Performance Drops in Visual Grounding Caused by Pruning. https://arxiv.org/abs/2506.21873
> >
> > [2] Token Pruning in Multimodal Large Language Models: Are We Solving the Right Problem? https://www.arxiv.org/abs/2502.11501
> >
> > [3] VScan: Rethinking Visual Token Reduction for Efficient Large Vision-Language Models. https://arxiv.org/abs/2505.22654

---

> > > ### Author Response · Authors · 2025-11-22
> > > **Response to Question 2, 3, and 4**
> > >
> > > > Q2:  Detailed ablation of token allocation strategy
> > > >
> > >
> > > A2: Thank you for this insightful question. In our experiments, when operating under a 50% total token budget, we first allocate a fixed proportion of tokens to IVC. The remaining tokens are then assigned to foreground tokens that carry semantic information relevant to the text prompt. As shown in Table 6 of our original submission, varying the proportion of IVC tokens reveals that allocating 10% to IVC is sufficient to retain spatial information without causing performance degradation. Under a 50% overall token budget, this choice leaves 40% of the budget for foreground tokens, which maximizes the capture of semantic cues.
> > >
> > > To further verify this allocation strategy, we conducted additional experiments on Qwen2.5-VL 7B, the results are summarized in Table R‑5. These experiments confirm that our configuration offers the best trade-off between spatial fidelity and semantic coverage. We have incorporated this ablation study and discussion into Appendix Section A.8 of the revised manuscript for completeness.
> > >
> > > Table R-5. Ablation study on token allocation under 50% budget (Qwen2.5‑VL 7B).
> > >
> > > | IVC Token Ratio | RefCOCO testA | RefCOCO+ testA | SeedBench | MMBench |
> > > | --- | --- | --- | --- | --- |
> > > | 5% | 91.8 | 87.2 | 76.6 | 82.5 |
> > > | 10% (Ours) | 92.0 | 87.4 | 76.7 | 82.6 |
> > > | 20% | 91.3 | 87.0 | 76.6 | 82.3 |
> > >
> > > > Q3:  Table 3 does not include data for PDrop.
> > > >
> > >
> > > A3: Thank you for this comment. We have included PDrop’s efficiency metrics in the revised comparison (Table R-6).
> > >
> > > In the original submission, we used FastV as the primary baseline because it represents the most computationally efficient attention-based pruning method. PDrop extends FastV with multi-stage progressive pruning, which requires computing attention scores at multiple intermediate layers rather than once at early layers. This design inherently introduces additional computational overhead under the same token retention ratio.
> > >
> > > To provide a complete comparison, we have now included PDrop’s efficiency metrics on Qwen2.5‑VL 7B. The revised comparison is shown in Table R‑6 below, and the corresponding update has been incorporated into Table 3 of the revised manuscript.
> > >
> > > Table R-6. Inference efficiency comparison on Qwen2.5‑VL 7B.
> > > | Method | Average Tokens (%)  | KV Cache (MB) ↓ | Prefill Time (ms) ↓ | Decode Latency (ms/token) ↓ | Total Time (mm'ss) ↓ | Accuracy (%) ↑ |
> > > | --- | --- | --- | --- | --- | --- | --- |
> > > | Vanilla | 100% | 26.0 (1.0×) | 408 (1.00×) | 65.3 (1.00×) | 60'17 (1.00×) | 92.2 |
> > > | FastV | 54% | 16.1 (1.6×) | 297 (1.37×) | 62.7 (1.04×) | 51'51 (1.16×) | 74.4 |
> > > | PDrop | 61% | 16.4 (1.6×) | 315 (1.30×) | 62.8 (1.04×) | 52'23 (1.15×) | 77.6 |
> > > | IVC-Prune | 50% | 15.9 (1.6×) | 322 (1.27×) | 60.2 (1.08×) | 47'47 (1.27×) | 92.0 |
> > >
> > > > Q4:  Table 5 does not include data for PDrop.
> > > >
> > >
> > > A4:  Thank you for this suggestion. We have conducted additional experiments to include PDrop across all model scales in Table R-7. Table 5 has been updated in the revised manuscript.
> > >
> > > Table R-7. Analysis of applying our method to Qwen2.5-VL models with different parameters.
> > >
> > > | Models | Method | RefCOCO testA | RefCOCO+ testA | SeedBench | MMBench |
> > > | --- | --- | --- | --- | --- | --- |
> > > | Qwen2.5-VL 3B | Vanilla |  89.6  | 82.5  | 73.8 |  76.7 |
> > > |  | FastV  | 81.2  | 71.0  | 70.3  | 73.8 |
> > > |  | *PDrop | 67.6 | 56.8 | 68.4 | 71.7 |
> > > |  | IVC-Prune  | 89.1  | 81.7  | 73.5  | 75.9 |
> > > | Qwen2.5-VL 7B | Vanilla | 92.2  | 88.0  | 76.7 |  82.4 |
> > > |  | FastV  | 74.4  | 68.9  | 72.9  | 80.5 |
> > > |  | PDrop | 77.6 | 72.1 | 74.0 | 78.9 |
> > > |  | IVC-Prune  | 92.0  | 87.4  | 76.7  | 82.6 |
> > > | Qwen2.5-VL 32B | Vanilla | 91.3  | 86.7  | 76.9  | 86.8 |
> > > |  | FastV  | 74.3  | 67.1  | 70.8  | 81.3 |
> > > |  | *PDrop | 49.8 | 43.6 | 66.0 | 68.0 |
> > > |  | IVC-Prune  | 91.1  | 86.3  | 76.7  | 85.8 |

---

> ### Author Response · Authors · 2025-11-22
> **Response to Question 5**
>
> > Q5: Additional results on high-resolution benchmarks such as DocVQA and OCRBench.
> >
>
> A5: Thank you for this valuable suggestion. Following the recommendation, we conducted supplementary experiments on three high‑resolution OCR‑related benchmarks: **DocVQA**, **InfoVQA**, and **OCRBench** using Qwen2.5‑VL 7B. The results are summarized in Table R‑8. On DocVQA and InfoVQA, IVC‑Prune achieves accuracy close to the Vanilla model and outperforms both FastV and PDrop, confirming its effectiveness in scenarios with a large number of visual tokens. On OCRBench, however, all pruning methods demonstrate notable performance degradation.
>
> Table R-8.  Results on high-resolution benchmarks (Qwen2.5-VL 7B).
>
> | Method | Average Tokens | DocVQA | InfoVQA | OCRBench |
> | --- | --- | --- | --- | --- |
> | Vanilla | 100% | 94.9 | 81.7 | 88.4 |
> | FastV | 52% | 93.9 | 76.4 | 67.3 |
> | PDrop | 52% | 93.8 | 69.4 | 62.0 |
> | IVC-Prune | 50% | 94.4 | 80.9 | 66.3 |
>
> To better understand the deterioration on OCRBench, we further analyzed performance across its task categories (Table R‑9). Our findings indicate that the **largest degradation occurs in recognition‑focused tasks**, in particular Text Recognition and Handwritten Mathematical Expression Recognition, while other VQA categories remain relatively robust.
>
> Table R-9. Task-level breakdown of OCRBench performance.
>
> | Task Category | Vanilla | IVC-Prune | FastV | PDrop |
> | --- | --- | --- | --- | --- |
> | Scene Text-centric VQA | 17.9 | 17.6 (98.3%) | 17.7 (98.9%) | 17.7 (98.9%) |
> | Doc-oriented VQA | 18.0 | 17.1 (95.0%) | 16.7 (92.8%) | 14.7 (81.7%) |
> | Key Information Extraction | 18.2 | 16.8 (92.3%) | 13.0 (71.4%) | 11.8 (64.8%) |
> | Text Recognition | 26.9 | 14.0 (52.0%) | 18.3 (68.0%) | 17.6 (65.4%) |
> | Handwritten Mathematical Expression Recognition | 7.4 | 0.8 (10.8%) | 1.6 (21.6%) | 0.2 (2.7%) |
> | Final Score | 88.4 | 66.3 (75.0%) | 67.3 (76.1%) | 62.0 (70.1%) |
>
> We attribute this gap to the nature of recognition-focused tasks: images typically contain densely packed characters or symbols with minimal background. Accurate recognition relies on preserving fine-grained visual details. Under a uniform 50% pruning ratio, a substantial portion of tokens encoding these details are removed, leading to inevitable performance loss.
>
> Visualization examples in Table 15 of the revised manuscript further support this analysis.  Failure cases originate from recognition-focused tasks and relatively low-resolution images. These inputs contain mostly foreground tokens relevant to text or symbolic content, leaving little redundant information to prune. In contrast, VQA-oriented inputs retain sufficient visual context even after pruning, sustaining strong performance.
>
> Overall, our findings highlight that the performance drop on OCRBench is not a general failure of the proposed IVC-Prune for high-resolution inputs, but rather a limitation in handling high-density text recognition. This suggests that adaptive or task-aware pruning strategies may be necessary to maintain accuracy in such scenarios.  A detailed analysis of this phenomenon is provided in Appendix Section A.6 of the revised manuscript.

---

### Meta-Review · Area_Chair_Nezw · 2026-01-11

**Summary:**

This paper was reviewed by four experts in the field. The recommendations are (4, 6, 6, 8). Based on the reviewers' feedback, the decision is to recommend the acceptance of the paper. The reviewers did raise some valuable concerns (especially more detailed experimental comparisons and ablation studies raised by Reviewer 9fb3, 4Zjm, and HRNj, clearer paper presentation and statement raised by Reviewer 4Zjm and HRNj, and more descriptions regarding methodological insight and motivation raised by Reviewer HRNj) that should be addressed in the final revised version of the paper. The authors are encouraged to make the necessary changes to the best of their ability.

**Reviewer Concerns:**

Through the rebuttal, the authors have successfully addressed several concerns regarding design motivation, paper presentation, and many experimental comparisons. However, some more in-depth analysis and discussion can be further elaborated to make the manuscript more convincing and stronger.

**Reviewer Scores:**

The reviewers may acknowledge and appreciate the authors' diligent efforts to improve the manuscript, particularly regarding the enhanced presentation, the clarification of the core motivation, and the inclusion of additional comparisons in the revised version. However, despite these positive steps, some remaining concerns persist regarding the depth of the analysis. The consensus is that the discussion requires further expansion to fully validate the findings and address the complexities of the proposed approach.

---

### Decision · Program_Chairs · 2026-01-26

Accept (Poster)